# The Metabolic Fates of Pyruvate in Normal and Neoplastic Cells

**DOI:** 10.3390/cells10040762

**Published:** 2021-03-30

**Authors:** Edward V. Prochownik, Huabo Wang

**Affiliations:** 1Division of Hematology/Oncology, UPMC Children’s Hospital of Pittsburgh, Pittsburgh, PA 15224, USA; huw14@pitt.edu; 2The Department of Microbiology and Molecular Genetics, UPMC, Pittsburgh, PA 15213, USA; 3The Hillman Cancer Center, UPMC, Pittsburgh, PA 15213, USA; 4The Pittsburgh Liver Research Center, Pittsburgh, PA 15260, USA

**Keywords:** anaplerosis, glutaminolysis, lactate dehydrogenase, malic enzyme, pyruvate carboxylase, TCA cycle, Warburg effect

## Abstract

Pyruvate occupies a central metabolic node by virtue of its position at the crossroads of glycolysis and the tricarboxylic acid (TCA) cycle and its production and fate being governed by numerous cell-intrinsic and extrinsic factors. The former includes the cell’s type, redox state, ATP content, metabolic requirements and the activities of other metabolic pathways. The latter include the extracellular oxygen concentration, pH and nutrient levels, which are in turn governed by the vascular supply. Within this context, we discuss the six pathways that influence pyruvate content and utilization: 1. The lactate dehydrogenase pathway that either converts excess pyruvate to lactate or that regenerates pyruvate from lactate for use as a fuel or biosynthetic substrate; 2. The alanine pathway that generates alanine and other amino acids; 3. The pyruvate dehydrogenase complex pathway that provides acetyl-CoA, the TCA cycle’s initial substrate; 4. The pyruvate carboxylase reaction that anaplerotically supplies oxaloacetate; 5. The malic enzyme pathway that also links glycolysis and the TCA cycle and generates NADPH to support lipid bio-synthesis; and 6. The acetate bio-synthetic pathway that converts pyruvate directly to acetate. The review discusses the mechanisms controlling these pathways, how they cross-talk and how they cooperate and are regulated to maximize growth and achieve metabolic and energetic harmony.

## 1. Introduction

### 1.1. General Features of Pyruvate Metabolism

In normal eukaryotic cells, and particularly during proliferative quiescence, oxidative phosphorylation (Oxphos) is the chief energy source and glucose is usually the primary fuel. Depending on the tissue, glucose is actively transported into cells by one or more of 14 different glucose transporters that comprise three distinct classes based on homology [1]. The anaerobic process of glycolysis then ultimately yields two molecules of pyruvate, two molecules of NADH and a net gain of two molecules of ATP for every molecule of glucose consumed. Upon diffusing across the mitochondria’s permeable outer membrane and then being actively transported across the impermeable inner membrane by the pyruvate carriers MPC1 and MPC2 and into the matrix [2], pyruvate is decarboxylated by the pyruvate dehydrogenase (PDH) complex (PDC). This yields an additional molecule of NADH and acetyl-coenzyme A (acetyl-CoA), the entry-level substrate for the TCA cycle. At maximal efficiency, the TCA cycle generates three molecules of NADH, a molecule of FADH2 and one molecule of GTP per molecule of pyruvate. Eventually, it generates an additional ~36 molecules of ATP per two molecules of pyruvate (or one molecule of glucose) oxidized [3]. The iterative oxidative regeneration of the TCA cycle’s intermediates via the continuous provision of acetyl-CoA reduces NAD^+^ and FAD^+^ and provides the electrons that are ultimately transferred to Complex I via NADH and Complex II via FADH2, respectively, and then to Complex III and Complex IV where molecular oxygen is reduced to water. The traversal of electrons through the electron transport chain (ETC), which is normally ~98% efficient [4], provides the requisite energy to pump protons from the matrix into the intermembrane space (IMS), thereby establishing an electrochemical gradient (also known as the mitochondrial membrane potential or ΔψM). The ΔψM is needed to generate ATP’s high-energy bond, as protons in the IMS re-enter the mitochondrial matrix via Complex V (ATP synthase) [5].

Alternate sources of acetyl-CoA are recruited and may even predominate during the periodic glucose deprivation that accompanies fasting, during the disposal of excess dietary lipids, during the catabolism of free branched-chain amino acids or in response to energy-demanding activities such as proliferation and/or the translation of highly secreted proteins. With the exception of the brain, organs with particularly high energy demands such as the heart and kidney normally prefer fatty acids as their primary energy source and skeletal muscles are intermittently dependent on beta-type fatty acid oxidation (β-FAO) in a manner that reflects their workload [6,7,8,9]. Acetate, whether ingested directly, synthesized by the gut microbiome, derived from various deacetylation reactions or generated from pyruvate itself, also provides a variable and controllable source of acetyl-CoA [10,11,12]. Under these conditions, or in response to the metabolic rewiring initiated by energy-demanding processes, pyruvate may be redirected for purposes other than supplying acetyl-CoA. Thus, poised as it is between the intersections of glycolysis, the TCA cycle and several biosynthetic pathways, pyruvate serves in a number of highly regulated capacities.

### 1.2. A Unique Role for Pyruvate in Embryogenesis

Pyruvate appears to play a unique role in embryogenesis where, beginning at the one-two cell stage, it is an essential substrate and remains so until late in the morula phase when glucose subsumes this role [13,14]. Indeed, pyruvate deprivation or PDC inhibition causes growth arrest and a decline of citrate and α-ketoglutarate levels at the two-cell stage [15]. Several TCA cycle enzymes that are critical for Oxphos, including PDC, are nuclearly localized during this period and pyruvate is needed to support the concurrent acetylation of histones H3K4Ac and H3K27Ac. PDC thus appears to serve as a direct nuclear source of acetyl- CoA needed for these post-translational modifications and the global epigenetic remodeling and zygotic gene activation that accompanies the earliest stages of embryogenesis [16]. However, the role of pyruvate as an actual energy source during this earliest time remains unknown.

### 1.3. Pyruvate Metabolism in Its Broader and Cross-Talking Context

This review discusses the six known fates of pyruvate with particular emphasis on how they are transiently altered or permanently deregulated to accommodate changes in the metabolism of normal and neoplastic cells (Figure 1). The reactions to be discussed below and the mechanisms underlying their control, include: 1. the lactate dehydrogenase (LDH) reaction, 2. alanine biosynthesis, 3. the PDC reaction, 4. the pyruvate carboxylase (PC) anaplerotic reaction, 5. the malic enzyme (ME) reaction and 6. acetate biosynthesis. Each is discussed in the context of other pathways, such as glycolysis, gluconeogenesis and glutaminolysis with which they share considerable regulatory cross-talk. Attempts to normalize or inhibit the aberrant pyruvate metabolism that often accompanies cancer or non-neoplastic disorders such as diabetes mellitus to achieve a selective therapeutic effect have generated considerable interest and are discussed only briefly since excellent in-depth discussions on this subject have recently appeared [17,18,19,20,21,22,23,24,25].

### 1.4. General Features of Pyruvate Generation from Glucose

Glycolysis fulfills at least five requirements of normal and neoplastic cells although the relative importance of these differs among cell types and depends upon their proliferative rates, their immediate energy-generating and biosynthetic rates and their environment. First and foremost glycolysis supplies ATP while maintaining a reduced cytosolic environment that is maintained by the NAD^+^/NADH balance. Second, glycolysis-derived pyruvate directly provides the TCA cycle with acetyl-CoA via PDC and/or oxaloacetate via PC and thus contributes to efficient energy generation via Oxphos. Third, during episodic hypoxia, the anaerobic process of glycolysis can increase over ten-fold to maintain ATP levels as happens in skeletal muscle during vigorous exercise [26,27]. Fourth, in rapidly proliferating normal and neoplastic cells, increased glycolysis not only supports ATP production but also provides glycolytic intermediates that are diverted into biosynthetic pathways to generate ribose sugars, amino acids such as glycine and serine and NADPH, which contributes to maintaining a reduced intracellular environment and further supports fatty acid synthesis (FAS) [28,29,30]. This support of anabolic activity, which may occur under either hypoxic or normoxic conditions, is known as the Warburg effect. When necessary, it can support the high biosynthetic demands of proliferating cells yet still provide sufficient pyruvate to sustain oxygen-dependent Oxphos [28,31,32,33,34,35,36,37,38]. Finally, glycolytically derived pyruvate can enter the TCA cycle to support both Oxphos and additional biosynthetic needs. How rapidly glucose is metabolized and the pathways utilized also differ greatly and are affected by the ambient oxygen tension of the extracellular milieu and, of course, the availability of glucose. Needless to say, because some of the above requirements may be of a transient nature and not always beneficial, plasticity and tight control of the glycolytic pathway are of paramount importance and must be both rapidly and efficiently executed. During maximal demand, however, the rate of glycolysis, which can increase as much as 100-fold, can vastly exceed the capacity for glucose to be fully oxidized via Oxphos [39].

## 2. The Lactate Dehydrogenase (LDH) Reaction and Alanine Biosynthetic Pathway

### 2.1. LDH Supports Glycolysis and Provides an Alternative Fuel Source

LDH is a critical and widely expressed cytoplasmic (and occasionally mitochondrial) enzyme that reversibly catalyzes the conversion of pyruvate to lactate and, in doing so, generates NAD^+^ [40]. This latter reaction is thus coupled to glycolysis as it recycles the electron acceptor needed to support this oxidative pathway.

LDH’s functional form is tetrameric and comprised of varying combinations of two LDH isoforms, designated M (LDH-A) and H (LDH-B). The A4/M4 homotetramer (also known as LDH5) predominates in skeletal muscle whereas the B4/H4 homotetramer (also known as LDH1) predominates in the heart. Other tissues contain variable proportions of each of the remaining three possible tetrameric forms and all five have somewhat different functions [41,42,43]. LDH-A tends to favor the production of lactate whereas LDH-B tends favors the production of pyruvate. For this reason, it is not surprising that many cancers demonstrate a selective over-expression of LDH-A and that the *LDHA* gene is a direct target of the c-Myc (Myc) oncoprotein, which also drives glycolysis [44,45,46]. However, the unidirectional substrate preference of LDH-A and LDH-B in only relative and the knockdown of either gene alone has little effect on lactate production or cell growth [47,48]. Due to this functional redundancy, only a double knockout of LDH-A and LDH-B causes a significant impairment of lactate production, NAD+ generation and Warburg effect and in cancer cells [49]. The resultant growth inhibition may also reflect considerably more than the cell-intrinsic ability to sustain glycolysis and provide anabolic substrates since high levels of extracellular lactate also contribute to such tumor-extrinsic advantages as immune evasion, angiogenesis and metastasis [50].

In exercising skeletal muscle where anaerobic glycolysis predominates and further oxidation of pyruvate by Oxphos is prohibited, the rapid LDH-A-mediated catalysis of pyruvate to lactate allows for the latter’s rapid excretion and entry into the circulation where it is converted back to pyruvate in the liver, largely via LDH-B. Pyruvate that is not then catalyzed to acetyl-CoA may be converted to oxaloacetate, where it enters the gluconeogenic pathway. This so-called lactic acid cycle or Cori cycle is important for maintaining plasma glucose levels, particularly during fasting when >90% of the glucose supply derives from gluconeogenesis and with more than a third of this being derived from the Cori cycle (Figure 2) [51,52,53,54]. In congenital LDH-A deficiency, the inability to convert pyruvate to lactate is associated with NADH accumulation, high levels of plasma pyruvate and exercise intolerance as a consequence of reduced skeletal muscle glycolysis, rhabdomyolysis and myoglobinuria [55]. In cancer, the repurposing of gluconeogenesis can provide glucose to the tumor microenvironment which, due to its often poorly vascularized nature, is often nutrient-depleted [56,57].

### 2.2. Gluconeogenesis Can Support Cancer Cell Survival and Growth

The role of lactate as a fuel in three non-small cell lung cancer (NSCLC) cell lines which differed in their *LKB1(STK11)*, *KRAD* and *TP53* mutational status was studied by Leithner et al. [58,59]. All three showed a net excretion of lactate when extracellular glucose concentrations were high and net lactate consumption when glucose concentrations were low. The expression and activity of mitochondrial phosphoenolpyruvate carboxykinase 2 (PCK2) was also inversely related to the extracellular glucose concention. In cells maintained under low glucose/high lactate conditions, exogenous ^13^C_3_-lactate was a significant source of PCK2-dependent phosphoenol pyruvate (PEP), the most proximal gluconeogenic substrate within the glycolytic pathway. High PCK2 expression was also identified immunohistochemically in 70% of primary lung cancer samples. These studies provided compelling evidence for the active participation of the gluconeogenic pathway in maintaining cancer cell proliferation. This initiates with the uptake of lactate followed by what is likely its LDH-B-mediated conversion to pyruvate, the PC-dependent generation of oxaloacetate and oxaloacetate entry into the gluconeogenic pathway via PCK2 and the generation of PEP.

### 2.3. The Direct Generation of Alanine from Pyruvate

During periods of energy-replete amino acid biosynthesis and translation such as occur during rapid proliferation, pyruvate is the immediate precursor of alanine. The generation of alanine occurs as the result of a transamination reaction between pyruvate and glutamate with the latter being the amino donor and generating α-ketoglutarate, which is then further processed as an anaplerotically-derived Oxphos substrate. This reaction is driven by alanine aminotransferase/glutamic-pyruvate transaminase (ALT/GPT). However, during more prolonged periods of starvation, a reversal of this biosynthetic process and deriving primarily from the breakdown of skeletal muscle proteins, can complement the Cori cycle so as to maintain a constant pyruvate supply [60,61].

## 3. The Pyruvate Dehydrogenase Complex (PDC)

### 3.1. PDC Is a Large, Multi-Subunit Complex

The strategies adopted by both normal and cancer cells to maximize the Warburg effect include the down-regulation of PDC, a large multi-component enzymatic complex that irreversibly catalyzes the intra-mitochondrial oxidative decarboxylation of pyruvate to acetyl-CoA and CO_2_, while generating a molecule of NADH. As a key link between glycolysis and the TCA cycle, the PDH activity of the PDC is key to determining the rate of pyruvate flux into the mitochondria as well as the rate of acetyl-CoA biosynthesis derived from pyruvate. It is therefore unique with regard to its “gatekeeper” function that bridges the major aerobic and anaerobic energy-generating pathways [42,62]. The NADH generated by PDC also provides one source of electrons to supply the ETC. A key aspect of PDC’s function is its ability to switch rapidly between active and inactivate states in a manner that is closely coupled with the cellular metabolic status and energy level [42,62].

The nearly 10 megadalton PDC consists of four components [63]: 1. PDH (the E1 subunit), which is a heteroterameric sub-complex comprised of two copies each of the E1α and E1β subunits, with each PDC complex containing 20–30 of these tetramers. 2. dihydrolipoyl acetyltransferase (the E2 subunit), which catalyzes the transfer of an acetyl group from a lipoate moiety to generate dihydrolipoate and acetyl-CoA. PDC contains 40–42 E2 subunits; 3. dihydrolipoyl dehydrogenase (the E3 homodimeric subunit), which regenerates lipoate from dihydrolipoate. PDC contains 6–12 E3 homodimeric E3 subunits; 4. the E3-binding protein (E3BP) component, with 18–20 members per PDC complex. It lacks any known enzymatic function, appears to serve a structural role and is essential for recruiting E3 to the complex [42,62,63].

### 3.2. PDC Regulation Is Highly Dependent on Metabolite Content, ATP Levels and Redox State

PDC regulation occurs primarily in response to the phosphorylation/dephosphorylation of E1α residues Ser_203_, Ser_264_ and Ser_271_ by four inhibitory PDH-kinases (PDK1-4) and by the stimulatory PDP1 and PDP2 phosphatases. In turn, the PDKs and PDPs are allosterically regulated by small molecules, specifically ATP, AMP, acetyl-CoA and NADH [42,62,64,65,66,67,68,69]. High ATP/ADP, acetyl-CoA/CoA and NADH/NAD^+^ ratios stimulate PDKs whereas low ratios stimulate PDPs. In this way, pyruvate’s conversion to acetyl-CoA is directly tied to pre-existing fuel supplies, energy levels and redox state. β-FAO also represses PDC both by its generation of acetyl-CoA and NADH and by the direct physical interaction between PDC and β-FAO pathway enzymes [33].

Consistent with the notion that PDC regulates the Warburg effect, PDHE1α is phosphorylated in some but not all cancers in a manner that correlates with PDK1 [70,71,72,73,74,75,76]. PDKs are also up-regulated by HIF1α in response to hypoxia [71,72,77,78]). However, whether PDC activity itself is up- or down-regulated appears to be highly dependent on the nature of the cancer and, just as importantly, probably reflects a particular tumor’s relative inherent dependence on Oxphos and glycolysis, its choice of energy-generating substrates and perhaps even regional and transient metabolic preferences. Tumors with an abundance of inactive PDH (and hence PDC) may be somewhat analogous to those in which the low-affinity M2 isoform of pyruvate kinase (PK) replaces the high-affinity PKM1 isoform, thereby allowing for the accumulation of PEP and other upstream glycolytic intermediates needed to support the Warburg effect and anabolic growth [79]. While the PK and PDH reactions are equally irreversible (ΔG^0^ = −7.5 kcal for each), a block at the level of PDH potentially affords greater energetic and metabolic versatility for several reasons. First, by allowing glycolysis to proceed to completion, it generates the two ATP molecules that contribute to the overall net positive energy balance of the pathway. This would be particularly important for executing the anabolic and energy-consuming reactions to which the Warburg effect is presumably dedicated. Second, as opposed to PEP, pyruvate’s accumulation further favors the Warburg effect given that its LDH-mediated conversion to lactate generates the NAD+ needed to sustain glycolysis. Third, as noted above, pyruvate contributes to the anabolic objectives of the Warburg effect by serving as the initial biosynthetic substrate for alanine and other amino acids and by providing the essential anaplerotic substrate oxaloacetate. This further supports aspartate biosynthesis and gluconeogenesis (see below). Finally, the failure to replace PKM1 with PKM2 may not necessarily limit the supply of PEP; it may still be generated from oxaloacetate via the proximal portion of the gluconeogenic pathway, particularly if PDC activity is reduced [58].

### 3.3. The Contribution of Oncogenic Signaling to PDC Regulation

The above-mentioned PDHE1α phosphorylation sites are not coequal and the factors and circumstances governing their modification can in some cases be quite distinct from those dictating the phosphorylation of other sites. Fan et al. [80] identified Tyr_301_ as such a target for multiple oncogenic tyrosine kinases (TKs) such as EGFR, Bcr-Abl and Jak2 and showed that phosphorylation blocked pyruvate binding and increased the Warburg effect. In contrast, the expression of a phosphorylation-defected mutant (Y301A) reduced aerobic glycolysis and increased Oxphos.

Ser_293_ is also highly modified (desphosphorylated) in hepatocellular carcinomas (HCCs) induced by Myc over-expression and in hepatoblastomas induced by the co-expression of mutant forms of β-catenin and yes-associated protein, which also dysregulate Myc [71,74,76,81,82] The resultant activation of PDH (and hence PDC) resulting from this modification was proposed to represent a means by which limiting amounts of pyruvate could be efficiently utilized by the TCA cycle to sustain both glycolysis and Oxphos in these rapidly growing tumors.

An additional layer of PDC regulatory complexity, independent of its phosphorylation and mediated by acetyl-CoA acetyltransferase (ACAT1), has been described in certain cancers [65]. Mitochondrial ACAT1 reversibly catalyzes the conversion of acetyl- CoA to acetoacetyl-CoA. ACAT1′s active form is a homo-tetramer whose structural integrity and activity are maintained by its Tyr_407_ phosphorylation mediated by the same TKs that phosphorylate Tyr_301_ of PDHE1α, including EGFR and FGFR1 [80]. The recognition of these sites by multiple TKs coupled with the fact that phosphorylation occurs in a variety of primary cancers and cancer cell lines [65,66,80,83] suggests that this is a common means of coordinately regulating PDH and ACAT1 activities. The fact that recombinant EGFR and FGFR1 phosphorylated purified recombinant ACAT1 at Tyr_407_ in vitro also proved that their phosphorylation of this residue was direct. Mutation of Tyr_407_ abolished both ACAT1′s tetramerization and enzymatic activation. The means by which membrane-bound or cytosolic TKs phosphorylated mitochondrial ACAT1 was explained by the finding that small amounts of these TKs reside in mitochondria [65,66,83]. However, the possibility that some ACAT1 was phosphorylated prior to its transport into mitochondria was not strictly ruled out [65,66,83]. Together, these findings indicate that the TK mediated phosphorylation-dependent activation of ACAT1 leads to inhibitory acetylation of PDHE1α via a mechanism that is independent of PDK/PDP-mediated phosphorylation/dephosphorylation. Despite the readily discernible phosphorylation of ACAT1 on Tyr_407_, by recombinant FGFR1 in vitro, neither it nor any of the other three putative phosphorylation sites appeared to be major phosphorylation site in cells. This suggested that low-level phosphorylation might be sufficient to exert all the observed effects and/or that efficient tetramerization of ACAT1 does not require the phosphorylation of all subunits.

The functional consequences of the above post-translational modifications were examined in human H1299 NSCLC cells in which endogenous ACAT1 was knocked down and then reconstituted with wild-type (WT) ACAT1 or the non-phosphorylatable Y407F mutant [65]. The latter cells proliferated normally under normoxic conditions but less well under hypoxic conditions or as tumor xenografts, suggesting that they were unable to generate sufficient pyruvate and other glycolytic intermediates via the Warburg effect. These cells also showed lower levels of glycolysis, lactate production and PDHE1α subunit phosphorylation and acetylation. Y407F-expressing cells were also more sensitive to Complex V inhibition by oligomycin, indicating that, lacking an efficient glycolytic switch, they became increasingly reliant on Oxphos. The hypoxia-specific proliferation defect of Y207F cells could be rescued by expressing the K202Q acetylation mimetic of PDP1, which, like PDP2, also activates PDC. By acetylating the PDHE1α subunit directly at Lys_321_, ACAT1 also directly inhibited PDC.

The above findings indicate that PDC activity is regulated by a two-step process. The first involves the tetramerization and oncogenic TK-mediated phosphorylation, stabilization and activation of ACAT1. The second step involves tetrameric ACAT1′s acetylation-mediated direct inactivation of the PDCα subunit as well as the indirect inhibition of PDC via PDP1 inactivation. Collectively, these reduce pyruvate’s flow into the TCA cycle and allow upstream glycolytic intermediates to accumulate for biosynthetic purposes while concurrently generating lactate in order to maintain NAD^+^ generation to support glycolysis. Oxphos activity might be further reduced by the loss of NADH that accompanies the PDC reaction.

### 3.4. The Therapeutic Targeting of PDH and Its Limits

Based on the same rationale that inhibiting PKM2 might represent an effective chemotherapeutic approach for certain cancers, some studies have reported success with the orally available small molecule pan-PDK inhibitor dichloroacetate (DCA), which increases PDC activity and reduces the pool of pent up Warburg intermediates [22,23,84,85]. This, plus the ensuing “normalization” of mitochondrial activity, disrupts tumor energetics and in some cases increases chemotherapeutic sensitivity most likely by restoring suppressed mitochondrial apoptotic pathways [79,86,87].

Not all tumors show evidence of PDC suppression. Indeed, our own work has shown that PDC activity is already low in normal murine liver, which relies more on β-FAO than glycolysis [71,73,88]. In the HCC and hepatoblastoma mouse models mentioned above [71,81,82], we observed a reversal of this metabolic dependency, It is therefore not immediately clear that increasing PDC activity even further with a drug such as DCA would necessarily impact the growth of such tumors.

### 3.5. Disrupting PDH Has Surprisingly Little Effect on Proliferation

Jackson et al. examined PDC’s role in normal and neoplastic hepatocte proliferation in mice bearing a hepatocyte-specific deletion of the PDHE1α subunit [88]. To examine normal proliferation, they employed a mouse model of Type I hereditary tyrosinemia in which animals lacking fumaryl acetoacetate hydrolase (FAH), the terminal enzyme in tyrosine catabolism, succumb to liver failure as they accumulate cytotoxic tyrosine catabolites [89] unless the more proximal enzyme p-hydroxyphenylpyruvate dioxygenase (HPD) is blocked by the drug 2-(2-nitro-4-trifluoro-methyl-benzoyl)-1,3-cyclo-hexanedione (NTBC). In a competitive repopulation assay, *pdhe1a^−/−^* and *pdhe1a^+/+^* hepatocytes were combined and co-injected in equal numbers into the spleens of recipient *fah^−/−^* mice, which were then slowly cycled on and off NTBC until achieving NTBC-independence as a result of donor hepatocyte expansion and replacement of diseased recipient hepatocytes. Under these conditions, *pdhe1a^−/−^* hepatocytes displayed no proliferative disadvantage, with the ratio of recovered, post-transplant donor *pdhe1a^−/−^* and *pdhe1a^+/+^* hepatocytes equaling the input ratio.

The consequences of PDH loss on hepatoblastoma growth differed depending on whether PDH was eliminated in all hepatocytes (pan-knockout [KO]) or only in those destined to become transformed (restricted-KO). In both KO groups, survival was modestly longer relative to that of tumor-bearing *pdhe1a^+/+^* control mice. Surprisingly however, pan-KO tumors were smaller than all others. Mice also had markedly elevated lactate levels and their demise was thought to be due to a combination of tumor burden and severe metabolic acidosis. Their tumors likely restricted diaphragmatic excursion and compromised respiratory compensation. These findings indicated that tumor-bearing *pdha1^+/+^* mice maintained normal serum lactate levels because excreted lactate was rapidly converted back to pyruvate by surrounding liver cells. Similarly, animals bearing restricted-KO tumor cells likely relied on the intact PDC activity of neighboring cells to metabolize lactate. Mice with pan-KO tumors re-metabolized lactate to pyruvate less efficiently in either transformed or normal hepatocytes and thus developed acidosis.

Other alterations in PDH-KO livers and tumors likely reflected the compensatory changes to the loss of pyruvate-derived acetyl-CoA Both PDH-KO livers and tumors had significantly reduced levels of acetyl-CoA and ATP. This probably explained the slower tumor growth rates but did not impede the proliferation of non-transformed hepatocytes. Second, consistent with their switch from β-FAO to aerobic glycolysis, tumors reduce their mitochondrial mass [73,76,88,90], which was less prominent in pan-KO tumors. This suggested that tumors compensate for PDH’s absence by maintaining mitochondrial mass to generate acetyl-CoA less efficiently and from less abundant alternative sources.

### 3.6. Disrupting PDH Alters the Steady State Metabolome, Redox-State and Acid-Base Balance

Metabolomic profiling of the above tissues by quantitative mass spectrometry (MS) confirmed the lower levels of acetyl-CoA in PDH-KO livers and tumors [88]. The former also contained lower levels of α-ketoglutarate, malate, NAD^+^ and NADH, which, combined with the previous findings, suggested that these mitochondria were engaged in a futile attempt to normalize TCA cycle activity and ATP production. Finally, mitochondrial citrate synthase (CS) activity was significantly higher in PDH-KO livers, in keeping with its inhibition by high ATP:ADP and NADH:NAD^+^ ratios [91]. CS, along with a plentiful supply of oxaloacetate provided by the PC reaction, might serve as an enzymatic “escape valve” to dispose of excess pyruvate that accumulates in the PDH-KO background. PDH-KO hepatoblastomass also contained lower levels of TCA cycle substrates compared to control tumors. Thus, the genetic dissociation of glycolysis from the TCA cycle and the ensuing suppression of ATP, acetyl-CoA and TCA cycle substrates impacted proliferation only modestly and only in highly proliferative tumors. Moreover, despite alterations in other metabolic pathways none were able to entirely replenish these essential products of glycolysis and Oxphos.

These findings were extended by Wang et al. to immortalized *pdhe1α^−/−^* rat fibroblasts which, along with WT cells, were engineered to conditionally over-express Myc [75]. Myc’s activation in WT cells induced PDC and rapid PDHE1α- dephosphorylation, as previously observed in Myc-driven HCCs [71,73,90]. Similar to PDH-KO hepatocytes [88], *pdhe1α^−/−^* fibroblast proliferation was equivalent to that of WT cells although they were ~15% smaller and accumulated more of the fluorescent glucose analog NBDG. Increased lactate production in *pdhe1α^−/−^* cells could be demonstrated, but only when they were grown to high densities and subsequently maintained under low serum conditions [75].

*pdhe1α^−/−^* fibroblast mitochondria displayed attenuated oxygen consumption rates (OCRs) in response to several TCA cycle substrates including pyruvate. They contained lower levels of NAD^+^ and NADH but higher NAD^+^:NADH ratios. Moreover, whereas WT fibroblasts increased NADH following Myc activation, *pdhe1α^−/−^* cells did not. This confirmed that some aspects of Oxphos are under Myc control in fibroblasts [31,32] and that the *pdhe1α^−/−^* cell defect was perhaps due to an inability to generate NADH from the PDH reaction, thus explaining the higher NAD^+^:NADH ratio. This, plus the reduced flow of protons into the mitochondrial matrix to generate ATP, might have explained the higher ΔψM of *pdhe1α^−/−^* cells. 

In addition to being more reliant on β-FAO [75,88], *pdhe1α^−/−^* fibroblasts showed more robust uptake of glutamine, exaggerated OCR responses to glutamate, increased mitochondrial mass and reduced reactive oxygen species (ROS) production. These results suggested that *pdhe1α^−/−^* cells’ failure to generate acetyl-CoA from pyruvate was compensated for by switching to β-FAO as seen in PDH-KO livers and hepatoblastomas [88]. *pdhe1α^−/−^* fibroblasts also increased glutaminolysis, which did not occur in PDH-KO livers or hepatoblastomas. Glutamine’s conversion to oxaloacetate is accompanied by the generation of two NADH molecules, one FADH2 molecule and one GTP molecule, all of which occur in an acetyl-CoA-independent manner via the “bottom half” of the TCA cycle independently of the acetyl-CoA-dependent “top half”. Increased glutaminolysis thus seems well-positioned to address the redox deficit of PDH-KO cells. The importance of glutaminolysis was further underscored by showing that *pdhe1α^−/−^* fibroblast viability was more dependent than WT cells on exogenous glutamine. Yet, the steady-state levels of glutamine, glutamate, α-ketoglutarate, succinate, fumarate and malate of the former were significantly lower whereas oxaloacetate and aspartate levels were higher. This suggested that the bottom half of the TCA cycle was accelerated to provide reducing equivalents in an acetyl-CoA-independent manner.

The above group also explored whether *pdhe1α^−/−^* fibroblasts’ glutamine dependency reflected the activation of an alternate pathway of lipid biosynthesis in response to acetyl-CoA pools that were insufficient to provide citrate from oxaloacetate and acetyl-CoA [75,92]. They asked whether citrate was furnished by reductive carboxylation of α-ketoglutarate [93,94,95,96] and ruled this out first by showing that isolated mitochondria retained their exaggerated response to glutamate indicating that it was independent of the cytoplasmic enzymes needed for lipid biosynthesis. Second, the pharmacologic inhibition of fatty acid synthesis exerted no effect on the glutamate response. Lastly, WT and *pdhe1α^−/−^* cells incorporated radio-labeled glutamine into lipids at similar rates, as well as in response to Myc induction, which normally increases de novo lipogenesis [46,71,96,97]. Wang et al. also found the cytoplasm of *pdhe1α1α^−/−^* cells to be more acidic and both the cytoplasm and mitochondrial matrix to be more oxidized [75]. The former was speculated to be a manifestation of *pdhe1α1α^−/−^* cells’ increased glycolytic rate and the accumulation of acidic glycolytic intermediates. Respiratory CO_2_ production, which accounts for 30–100% of cellular acidification depending on whether energy is generated from glucose or fatty acids, may also have contributed [98]. The oxidized mitochondrial matrix of *pdhe1α^−/−^* fibroblasts likely resulted from the loss of PDH’s ability to reduce NAD^+^. The oxidized cytoplasm may have resulted from the slightly higher rate of NAD^+^-generating LDH-catalyzed lactate production which, along with an increase in PC activity, diverted pyruvate elsewhere. 

*pdhe1α^−/−^* fibroblasts showed markedly higher steady-state levels of numerous metabolites relative to WT cells. Among the top ten were six glycolytic intermediates, including pyruvate itself (6.4-fold), glucose (6.0-fold) and fructose 1,6,-diphosphate (4.6-fold). In response to Myc induction, WT cells increased pyruvate levels by 5.0-fold whereas those in *pdhe1α^−/−^* cells showed no further increase, most likely because they were already maximally reliant on glycolysis. Consistent with earlier observations that *pdhe1α^−/−^* cells increased lactate only under extreme conditions, basal levels in these cells were identical to WT cell levels. Together, these studies showed that down-stream blocks to glycolysis, whether they be via the replacement of PKM1 by PKM2, the deletion of *pdhe1α* or the phosphorylation-mediate silencing of PDHE1α via, promote the accumulation of glycolytic intermediates that are responsible for the pro-anabolic features of the Warburg effect.

*pdhe1α^−/−^* cells also contained high levels of fructose, mannose 6-phosphate and sorbitol/mannitol (5.4-fold, 4.4-fold and 1.2-fold increases, respectively). This suggested that glycolytic intermediate accumulation drove a reversal of the catabolic pathways utilized by non-glucose hexose sugars, which connect to the glycolytic pathway. This was supported by the finding that the fructose precursor dihydroxyacetone phosphate (DHAP) was the least altered of all glycolytic intermediates and nearly three-fold less abundant than its immediate precursor fructose 1,6-biphosphate. DHAP is a precursor of triglycerides and membrane lipids, the biosynthetic pathways for which are quite active in fibroblasts [99,100,101]. Indeed many of the 173 measured intermediates lying along these pathways were elevated in *pdhe1α^−/−^* fibroblasts. Indeed, Wang et al. [75] found that *pdhe1α^−/−^* fibroblasts incorporated more acetate and palmitate into lipids. The finding that less TCA cycle-generated acetyl-CoA and citrate were devoted to de novo lipid synthesis, that WT and *pdhe1α^−/−^* cell mitochondria demonstrated equivalent levels of β-FAO and that reducing equivalents in the latter were generated via glutaminolysis in an acetyl- CoA-independent manner, likely explained why they and WT cells contained equivalent levels of acetyl-CoA.

Other than glutamine and glutamate, the most depleted amino acids in *pdhe1α^−/−^* fibroblasts were proline and arginine, both of which are derived from glutamate. This, coupled with the five-fold higher levels of oxaloacetate in *pdhe1α^−/−^* cells and the 1.5-fold higher levels of aspartate (the only amino acid whose levels were increased) further supported the idea of increased flux across the bottom of the TCA cycle. 

Finally, Wang et al. [75] used U [^15^N,^13^C]-glutamine labeling and quantitative MS to measure isotopic incorporation into glutamate, citrate, succinate, malate and aspartate. Consistent with their proposed higher rates of glutamine flux, *pdhe1α^−/−^* cells contained proportionately more uniformly labeled glutamate and less unlabeled glutamate. Isotopically labeled succinate, malate and citrate were also more abundant indicating their more rapid derivation from glutamate. Glutamine-derived aspartate also accumulated more rapidly in *pdhe1α^−/−^* cells, indicating that it originated from a higher rate of synthesis rather than reduced utilization. Finally, the disproportionately higher level of M+3 citrate in *pdhe1α^−/−^* cells supported its origin from malic enzyme–mediated conversion of M+4 malate to pyruvate, which was then converted to oxaloacetate by PC and then to (M+3) citrate.

## 4. Pyruvate Carboxylase (PC)

### 4.1. PC Is a Major Anaplerotic Source of Oxaloacetate

PC is a biotin- and ATP-dependent mitochondrial carboxylase that catalyzes the reversible mitochondrial conversion of pyruvate and CO_2_ to oxaloacetate [102,103,104,105]. PC, malic enzyme (ME-see Section 5) and PDC are the only means by which pyruvate directly supplies a primary TCA cycle intermediate (Figure 1). Moreover, the substrates generated by these are each precursors for citrate and oxaloacetate is a substrate for several non-TCA cycle-related pathways. Thus, these activities must be balanced and regulated in ways that allow the proper apportioning of these various substrates according to the needs of the cell, the source of the pyruvate and environmental circumstances.

During both quiescence and proliferation, certain TCA cycle intermediates exit the mitochondria (cataplerosis) where they may be converted back to glucose (gluconeogenesis) or used for the biosynthesis of amino acids, lipids and nucleotides. However, this runs the risk of depleting and/or slowing, the TCA cycle and impairing ATP generation, particularly during rapid proliferation. Replenishing these substrates by converting pyruvate to oxaloacetate and converting glutamine and glutamate to α-ketoglutarate [103,106,107,108,109], are the principal means of ensuring a sustainable balance between anabolism and energy generation.

A classic anaplerotic challenge develops when citrate enters the cytoplasm and is converted to oxaloacetate and acetyl-CoA via the action of ATP citrate lyase (ACLY). Oxaloacetate may participate in gluconeogenesis and the biosynthesis of aspartate, asparagine and nucleotides whereas acetyl-CoA is the foundational substrate for both lipid biosynthesis [92,102]. The ACLY reaction can indirectly influence the pyruvate supply given that cytoplasmic citrate is a potent inhibitor of phosphofructokinase (PFK), one of three rate-limiting glycolytic enzymes [8]. Similarly, succinyl-CoA and fumarate can be diverted from the TCA cycle for the biosynthesis of select amino acids. Here, too, cells rely heavily upon pyruvate anaplerosis to replace these depleted substrates and thus to maintain the balance necessary for idealizing energy generation while addressing cataplerotic needs and their attendant anabolic energy demands. PC-mediated conversion to oxaloacetate, along with glutaminolysis, represent the two most prominent routes of mitochondrial anaplerosis [25,102,106,110,111,112,113,114,115].

Although glucose and glutamine are the most abundant plasma nutrients, their concentrations in the tumor micro-environment may be 10-40 times lower owing to regional differences in blood supply [57,104,116,117,118,119,120]. It is important for both normal and tumor cell survival that glucose and glutamine be utilized not only efficiently but coordinately so as to permit maximum flexibility in response to fluctuations in their supply, which can occur abruptly. The mitochondrial fate of pyruvate can be fine-tuned to furnish either acetyl-CoA via PDC or oxaloacetate via PC in order to balance anaplerotic demand with oxygen availability. Indeed, in certain tissues such as the liver, up to four times more pyruvate is converted to oxaloacetate than to acetyl-CoA since acetyl-CoA can be derived from multiple sources [114]. α-ketoglutarate, the TCA cycle endpoint of glutaminolysis can be used to provide the next downstream TCA cycle substrate (succinyl-CoA) or can undergo reductive carboxylation to citrate, thus providing a non-glycolytic source of citrate and additional extra-mitochondrial acetyl-CoA [93,95].

Aside from participating in protein biosynthesis, glutamine and oxaloacetate-derived aspartate fulfill additional anabolic roles as nitrogen donors during the biosynthesis of other amino acids, nucleotides, amino sugars and NAD(P)^+^. Not surprisingly, Myc directly stimulates both PC anaplerosis and glutaminolysis as well as glycolysis, Oxphos and other anabolic pathways [28,31,32,96,109,111,121,122,123,124,125].

### 4.2. Anaplerotically Derived Oxaloacetate Contributes to Multiple, Non-Mitochondrial Anabolic Pathways

PC-generated oxaloacetate contributes to several metabolic pathways including gluconeogenesis and the synthesis of glycogen, fatty acids, cholesterol, aspartate and asparagine. For gluconeogenesis, oxaloacetate, which cannot cross the inner mitochondrial membrane, must first be reduced to malate by mitochondrial malate dehydrogenase (MDH). Malate is then transported to the cytoplasm where it is re-oxidized to oxaloacetate by cytoplasmic MDH before entering the gluconeogenic pathway and being converted to PEP in a GTP-dependent reaction catalyzed by PCK. This pathway, particularly active in the liver, kidney cortex and pancreatic β-cells, is critical for maintaining blood glucose levels during periods of starvation or dietary carbohydrate inadequacy. In such circumstances, pyruvate for the PC reaction originates from sources other than glucose and may include lactate or gluconeogenic amino acids. This pathway is activated not only as a result of starvation-mediated hypoglycemia but also in response to tumor-associated cachexia, inflammation and elaboration of cytokines and cytokine receptors [126,127,128,129,130]. Inherited deficiencies of PC are associated with lactic acidosis and hypoglycemia, particularly during fasting [78,104].

While generally regarded as being confined to the liver and renal cortex, gluconeogenesis is hijacked and up-regulated in some cancers [131]. Hepatic gluconeogenesis is also increased in animals bearing non-hepatic tumors through poorly understood mechanisms. For example, Liu et al. measured the flux of [3-^13^C]alanine into pyruvate and other metabolites in the livers of rats bearing mammary tumors and demonstrated increases in the PC-driven synthesis of oxaloacetate [132]. This suggested that hepatic generation of glucose in a non-neoplastic tissue may be mobilized to satisfy the nutritional demands of a distant neoplasm. However, Liu et al. [132] neither investigated whether increased hepatic gluconeogenesis actually benefited tumor growth nor explored how gluconeogenesis was up-regulated.

Of the 11 enzymatic steps in gluconeogenesis, four, including the first (PCK), are gluconeogenesis-specific whereas the remaining seven involve a reversal of glycolytic reactions. As gluconeogenesis occurs in numerous cancers [34,58,59,99,133], it might appear that the reversal of glycolysis during gluconeogenesis would antagonize the glucose flux and anabolic advantage provided by the Warburg effect. In fact, gluconeogenesis might be particularly beneficial when extracellular glucose concentrations are so limiting that even small amounts of glycolytic intermediates are better than none at all or when the glucose is used by a nearby cell, with those actually engaged in gluconeogenesis deriving energy from other sources [34,131].

Despite removing oxaloacetate from the TCA cycle, PCK can actually accelerate Oxphos by generating GDP, which is a co-factor for the succinyl-CoA synthase reaction that generates succinate from succinyl-CoA [134]. PCK is particularly important for tumor growth because, even in the absence of the other gluconeogenesis-specific enzymes, it generates PEP, a precursor for the tumor-critical anabolic substrates serine and glycerol 3-phosphate. PCK can also regenerate pyruvate in glucose-starved cells and participate in acetyl-CoA generation [133,135]. Many tumors thus up-regulate PCK given its capacity to furnish critical anabolic substrates, normally derived from glucose or glutamine anaplerosis, in an acetyl-CoA-independent manner [34,133,135,136]. In contrast, in cancers arising in tissues with a functionally intact gluconeogenic pathway, PCK functions as a tumor suppressor by tempering the Warburg effect [131] as mentioned above.

PC’s ability to furnish oxaloacetate for purposes other than gluconeogenesis varies widely. For example, PC levels in pancreatic β-cells rival those in the liver and kidney, with as much as half the pyruvate being converted to oxaloacetate [137,138,139]. As β-cells lack the gluconeogenic enzymes fructose-1,6-bisphosphatase and PCK [140,141,142] their pyruvate-derived oxaloacetate plays a disproportionate role in maintaining Oxphos [137,139]. Indeed, a link between PC and insulin secretion is suggested by studies showing that *PC* gene polymorphisms in African Americans, which determine its level of expression, are associated with differential responses to insulin and that the development of Type II diabetes is associated with reduced β-cell PC expression [138,139,143]. Dose-dependent correlations have also been noted in vitro between PC activity and insulin secretion rates in response to glucose and other secretagogues. Robust suppression of PC in β-cells markedly inhibited insulin secretion without affecting insulin content or glucose oxidation [137,139]. The depletion of citrate and, to a lesser extent, aspartate, likely reflected their lower rates of synthesis from PC-derived oxaloacetate. These results suggested that insulin secretagogues are not only reliant on PC but share a common mechanism of action involving TCA cycle maintenance, particularly the supply of glutamine-derived α-ketoglutarate and the reduced diversion of citrate for lipogenesis. Despite the above changes in TCA cycle substrate balance in response to PC suppression, the precise nature of the changes leading to reduced insulin section were not entirely established. By avoiding pyruvate’s oxidation to acetyl-CoA via PDC, PC supplies the TCA cycle in a manner than avoids the generation of NADH thus altering redox balance in ways that might also impact insulin secretion. Finally, excess PC-derived oxaloacetate and its reverse conversion to malate could potentially activate mitochondrial malic enzyme 2 ME2, thus allowing malate to regenerate pyruvate while producing additional reducing equivalents. In pancreatic β-cells, this PC/ME-dependent circuit is capable of producing more NADPH than the pentose phosphate pathway [144,145,146].

Other studies in INS-1 rat insulinoma and primary pancreatic β cells showed that in addition to reducing glucose-sensitive insulin release, PC suppression also inhibited proliferation whereas PC over-expression had the opposite effect [139]. This seemingly unrelated link between insulin secretion and proliferation is perhaps unsurprising when considering the need to maintain maximal TCA cycle function to support the energy-demanding tasks of insulin storage, secretion and re-synthesis. Insulin normally accounts for ~10–50% of β-cell protein, with 15–25% of this being secreted daily, although this can increase in the face of insulin resistance or excessive glucose intake [147,148,149]. The fact that β-cells convert virtually none of their pyruvate to lactate but nearly half of it to oxaloacetate [139,150] suggests that pyruvate is selectively diverted for Oxphos, with the acetyl-CoA supply likely deriving from the remaining pyruvate and/or β-FAO [151]. Indeed, PC inhibition in the above INS-1 cells reduced ATP levels by ~60% [139]. Although the status of AMP-activated protein kinase (AMPK) was not determined in this study, it seems likely that it should have been induced in response to this marked ATP depletion and would have contributed to the suppression of other energy-demanding processes while attempting to restore ATP levels [139,152].

### 4.3. Dysregulation of PC and Cross-Talk with Glutaminolysis Is Common in Tumor Cells

Using RNAseq data from The Cancer Genome Atlas (TCGA), we examined the relationship between PC transcript expression and survival in >10,000 tumors comprising 34 cancer types and identified five examples of associations between high PC expression and favorable survival (Figure 3). One interpretation of these findings is that high PC reduces the Warburg effect by preventing the accumulation of glycolytic intermediates [132,153,154]. Indeed, high rates of glycolysis for most of these cancers have previously been shown to be associated with inferior survival [155,156,157,158]. Glutaminolysis is a PC-, acetyl-CoA- and glucose-independent alternate source for oxaloacetate. Although glutamine is further removed from oxaloacetate in the TCA cycle than is pyruvate, many transformed cell lines acquire up to 90% of their oxaloacetate from glutamine, with some of this being derived via reductive carboxylation [106,159].

Glutaminolysis and the PC reaction cross-talk in ways that are highly tissue-dependent [104,116]. For example, Cheng at al studied metabolic reprogramming in two glutamine-dependent glioblastoma (GBM) cell lines in response to glutaminase knockdown [116]. This was accompanied by a reduction in the amount of ^13^C-labeled malate and citrate derived from ^13^C-glutamine and by an increased amount of label derived from D-[1,6-^13^C]-glucose or [1-^13^C]-pyruvate with only modest effects on proliferation. Interpreting these findings as evidence for a compensatory shift to PC-mediated anaplerosis, Cheng et al. [116] next derived a glutamine-independent cell line. Although no differences were noted in glucose utilization between this and the parent line, glutamine-independent cells expressed higher PC levels and transferred more label from D-[1,6-^13^C]-glucose or [1-^13^C]-pyruvate into citrate. This PC dependency was confirmed by showing that PC knockdown reduced growth in a manner that could be rescued by exogenous glutamine.

HepG2 HCC cells were also found by Cheng et al. to be reliant on glutamine anaplerosis and to furnish very little glucose- or pyruvate-derived ^13^C for the TCA cycle whereas the reverse was true for Huh-7 HCC cells [116]. These studies provided another example of adaptable cross-talk between the pyruvate-derived supply of oxaloacetate on the one hand and the glutamine-derived source on the other. The communication between these pathways was particularly prominent in glutamine-addicted cells, which could overcome this dependency by increasing PC activity. Cheng et al. [116] did not report whether PDC activity was similarly altered, which might have allowed for better-coordinated supplies of pyruvate-derived oxaloacetate and acetyl-CoA. Nonetheless, it serves as a cautionary reminder that targeting glutamine addiction therapeutically in cancer might be difficult to achieve, let alone maintain [113].Indeed, even the near-total suppression of glutaminase did not have particularly drastic consequences and might reflect the bypass of this reaction and the anaplerotic use of glutamate.

A subsequent study examined PC behavior directly in patients with NSCLC by monitoring PC protein levels and ^13^C_6_-glucose and ^13^C_5_,^15^N_2_-glutamine flux in 86 tumors and matched adjacent normal lung tissue [160]. Median tumor PC protein levels were seven times higher compared to normal tissues whereas glutaminase levels were equivalent. In 34 patients infused with ^13^C^6^-glucose prior to tumor resection, gas-chromatography-MS (GC-MS) showed the flux of label through the PC and PDC pathways to be about two-fold higher in tumor tissues. In 13 paired tumor and non-tumor samples cultured in ^13^C_6_-glucose- or ^13^C_5_,^15^N_2_-glutamine-containing medium, more ^13^C_3_-lactate and ^13^C_4_-, ^13^C_5_- and ^13^C_6_-labeled citrate were produced by tumors. As these latter isotopologs require several turns of the TCA cycle plus the combined actions of PC and PDC, it was concluded that both enzymatic activities were elevated in tumors. shRNA-mediated suppression of PC in several NSCLC cell lines and A549 lung adenocarcinoma cells promoted altered morphology, increased cell size, multi-nucleation, lower proliferation and clonogenicity, and, at least in the latter case, reduced xenograft growth.

Evidence for altered PC anaplerosis was further documented in more than 70% of low-grade gliomas harboring point mutations in the isocitrate dehydrogenase 1 (*IDH1*) gene that typically involve the Arg_132_ residue [161]. IDH1 catalyzes the oxidation of isocitrate to α-ketoglutarate but mutant IDH catalyzes the synthesis of the neomorphic onco-metabolite 2-hydroxyglutarate (2-HG) from α-ketoglutarate [162]. The reaction thus depletes α-ketoglutarate that would otherwise contribute to the TCA-cycle and thus compromises anaplerotic glutamine flux. Relying on immortalized human astrocytes expressing IDH1^WT^ or IDH1^R132^, Izquierdo-Garcia et al. [112] traced the fate of 2-^13^C-glucose and demonstrated in the latter cells increased flux of the label through the PC pathway and decreased flux through the PDH pathway, both of which correlated with enzymatic activities. Higher PC activity in IDH1^R132H^ cells also correlated with increased protein level whereas lower PDH activity correlated with inhibitory phosphorylation of PDHE1α, indicating differences in the control of each reaction. These results agreed with PC expression levels for both low- and high-grade human gliomas from TCGA [112]. Collectively, the results indicated that the compromise of glutaminolysis in IDH1 mutant gliomas is compensated for by increased, PC-mediated anaplerotic pyruvate flux and reduced PDC flux. This study did not determine whether the resulting stoichiometic disparity between pyruvate-derived oxaloacetate and acetyl-CoA was re-balanced by acetyl-CoA derived from non-glucose-derived sources.

### 4.4. PC Reprogramming Is Opportunistic and Benefits Both Aggressive and Metastatic Tumors

The degree to which PC- and glutamine-dependent anaplerosis demonstrate reciprocity is not fixed and can be altered by environmental changes [25,116,123,160,163,164,165]. Christen et al. examined the in vivo flux of ^13^C_6_-glucose into TCA cycle intermediates of primary tumors and pulmonary metastases of 4T1 murine breast cancer [166]. Based on the distribution of ^13^C in M+3 malate and M+3 succinate, the studies suggested that switching from low- to high-level PC-dependent anaplerosis is dictated by changes in metabolic needs, tumor growth rates and the environment. Both PC transcripts and pyruvate levels were increased in 4T1 lung metastases in association with a borderline decrease in MCT1. In in vitro studies performed on three human breast cancer cell lines, the addition of pyruvate increased its intracellular levels by ~10-fold and also increased PC activity as measured by ^13^C_6_-glucose flux [166,167]. This suggested that the switch from glutamine- to pyruvate-driven anaplerosis was dictated by local concentration differences in pyruvate and that the preference of metastatic tumors for pyruvate was adaptive and microenvironmentally determined [168].

Comparable findings were made by Phannasil et al. [169] who examined 57 primary human breast cancers and demonstrated that PC levels directly correlated with tumor size and stage but not with hormone receptor or Her2 status. They also studied ^13^C_6_-glucose and ^13^C_5_-glucose flux in MDA-MB-231 breast cancer clones with variable levels of PC knockdown and demonstrated reduced incorporation of label into TCA cycle intermediates and a direct correlation between PC activivity and proliferation [169]. Decreased glycolysis was noted in knockdown clones and correlated with reduced labeling of pentose phosphate precursors, glycine and serine. That ADP, ATP or NAD(H) levels were not affected led the authors to conclude that PC suppression did not impact mitochondrial energy production. However, an alternate explanation is that energy production was reduced but offset by slowed proliferation. Shinde et al. [170] further added to PC’s role in advanced and/or aggressive breast cancer by documenting PC gene ampification in nearly one-third of primary breast cancers from the MCTI and METABRIC data bases and further demonstrating an inverse correlation between PC expression and survival [170]. Other studies documented similar correlations between PC expression and proliferation in GBM, paraganglioneuroma and papillary thyroid, kidey and NSCLCs [160,171,172,173].

### 4.5. PC Helps to Maintain the Tumor Micro-Environment

Poor vascularization and high metabolic rates often leave tumors deprived of essential nutrients [57,174,175,176,177,178]. Tumor-derived lactate acidifies the extracellular space, which is compounded by declining oxygen levels and CO_2_ accumulation [57,115,118,165,179,180]. Under these conditions, tumor extracellular lactate:glucose ratios may approach 300 (versus the ~0.15 of normal serum) and the pH may fall below 6 [57,179,181]. PC may play a central role in normalizing this environment while providing much-needed fuel for aerobic or anaerobic glycolysis. This is achieved by the LDH-B-mediated conversion of lactate back to pyruvate, which the enters the gluconeogenesis pathway via PC [180,182]. Many tumor cells and tumor stem cells also up-regulate monocarboxylate transporters, particularly MCT1 and MCT4, which facilitate lactate uptake from the extracellular space [183,184,185,186,187,188,189]. This is particularly notable in the case of MCT4, which is more commonly over-expressed in cancer than is MCT1 and has a preferential affinity for lacate over pyruvate [190,191]. Indeed, MCT1 and MCT 4 appear to complement one another’s function to the ultimate benefit of different cancer cell populations residing in neighboring hypoxic and normoxic compartments. In the former case, MCT 4 preferentially transports lactate out of hypoxic (glycolytic) cells. It is then internalized by MCT1 by cells in normoxic niches, where, following re-conversion to pyruvate, it enters the TCA cycle [180,191]. This spares glucose, which can then be dedictated to supporting the hypoxic population via anaerobic glycolysis. Neither situation is ideal since the oxidative cells are deprived of the glucose needed to support the Warburg effect and the hypoxic cells generate potentially harmful lactate and are growth-inhibited. However, both populatons are nonetheless saved from anoxic- or starvation-based death. NADH generation during lactate’s conversion to pyruvate deprives the oxidative glycolytic pathway of essential NAD+, and reduces glycolysis and lactate production.

### 4.6. PC Reprogramming Can Impact the Response to Chemotherapy

Several studies have documented how changes in PC impact chemotherapy. Delgado-Goñi et al. [192] investigated the basis for acquired resistance to the B-Raf inhibitor vemurafenib in human A375 melanoma cells bearing the common oncogenic B-Raf^V600E^ mutation. They generated three vemurafinib-resistant clones, all of which up-regulated the MAPK pathway as commonly happens [193]. One clone showed decreased Glut1 transporter expression, lower glucose uptake and lactate production and a switch to Oxphos. Consistent with their less prominent Warburg effect, these cells proliferated slower than vemurafenib-sensitive cells and were less reliant on pyruvate, glutamine and glucose, despite up-regulating PDC. Anaplerotic pathway re-programming was evidenced by decreased glutaminolysis and increased PC expression. These findings were consistent with prior studies showing that chemotherapy-resistant B-Raf mutant melanomas often down-regulate glycolysis and up-regulate Oxphos anaplerosis [194].

The up-regulation of PC in Myc-over-expressing expressing cells [169,170,195] was confirmed by Lao-On et al. [109] who showed the *PC* gene to be a direct Myc target in highly aggressive MDA-MB-231 breast cancer cells. PC expression declined following genetic or pharmacologic knockdown of Myc. As expected both Myc inhibition and PC suppression suppressed proliferation, invasion and migration. Myc bound directly to two sites residing 291–471 bp upstream of the transcriptional start site of *PC*s’s P2 promoter. Mutation of either site reduced expression of a reporter gene by 50–60% and mutation of both reduced expression by 75%. Interestingly, these binding sites seemed to be important only in MDA-MB-231 cells but not in MCF7 cells which express less Myc. This suggested that these sites were low-affinity and could further explain how low levels of Myc selectively regulate glycolytic genes whereas higher levels also impact Oxphos, thus simultaneously maximizing both energy generation and the anabolic substrate supply. The finding that PC and glutaminolysis are directly regulated by Myc [109,111] unifies the two major pathways of mitochondrial anaplerosis under the roof of a single master transcriptional regulator that impacts multiple metabolic pathways as well as mitochondrial structure and function [28,31,32,110,122].

### 4.7. PC Reprogramming Rescues Tumor Cells with Defective Mitochondrial Function

Providing oxaloacetate to support gluconeogenesis is not PC’s only role. Cardaci et al. [171] studied the fates of pyruvate and oxaloacetate in non-transformed murine kidney cells lacking SDHB, one of the four subunits of succinate dehydrogenase (SDH), otherwise known as ETC Complex II. Mutations in SDH subunits, particularlly SDHB, occur in familial pheochromocytoma, paraganglioneuroma and other rare neoplasms leading to succinate accumulation and fumarate depletion [196,197,198,199,200]. This activates hypoxia- inducible factors, which inhibit α-ketoglutarate-dependent histone and DNA demethylases and promote global pseudo-hypoxia and hypermethylation that drive transformation [201,202,203]. *sdhb*-KO cells had ~20-fold higher basal levels of succinate and ~15-fold lower levels of fumarate, with both metabolites being mostly glutamine-derived. Pyruvate’s conversion to acetyl-CoA and the generation of citrate and lipogenic acetyl-CoA were also markedly reduced in *sdhb-*KO cells even though ATP generation increased. Perhaps as a compensatory response, *sdhb-*KO cells displayed increased mitochondrial mass and glycolytic rate, used exogenous pyruvate more efficiently and produced more lactate, all of which suggested a need to regenerate NAD^+^ to support glycolysis. Consistent with this, *sdhb-*KO cells showed a reduced NAD^+^:NADH ratio in response to pyruvate deprivation and accumulated glycolytic substrates upstream of the NAD^+^-dependent step in glycolysis involving the conversion of glyceraldehyde-3-phosphate to 1,3-bisphosphoglycerate. *sdhb-*KO cell growth was also dependent upon an exogenous source of pyruvate. Further metabolic profiling indicated that *sdhb-*KO cells were asparate-deficient, which was confirmed in primary human pheochromatcytomas and paragangliomas. Moreover, sustaining even this lower level of aspartate was dependent on exogenous pyruvate or aspartate. As ~80% of the aspartate in WT cells was derived from glutamine, they remained relatively unaffected by pyruvate withdrawal whereas *sdhb-*KO cells required both glutamine and pyruvate. The above results were confirmed in *SDHB*-wild-type and mutant human pheochromacytomas/parangliomas.

The foregoing findings indicated that PC anaplerosis can compensate for glutamine anplerosis when the bottom half of the TCA cycle is non-functional due to SDH dysfunction. Similar observarions have been made in other tumor types with IDH or fumarate hydratase mutations [95,112,204].

### 4.8. Cells with Defective Mitochondria Require Pyruvate-Derived Aspartate

The obvious question raised by the above-described studies pertains to the relationship between pyruvate- and glutamine-derived oxaloacetate and aspartate biosynthesis. As plasma aspartate levels are ~100 times lower than those of glutamine, it must be continuously synthesized using mitochondrially derived oxaloacetate. However, the above studies do not address aspartate’s role in regulating mitochondrial function. 

ETC defects impair proliferation in ways that can be rescued by supra-physiologic amounts of pyruvate [205,206,207,208]. In fact, pyruvate can even normalize the proliferation of so-called ρ^0^ cells, which are devoid of mitochondria [206]. Such pyruvate auxotrophy is surprising given that cells with defective Oxphos usually up-regulate glycolysis. Exogenous pyruvate was therefore hypothesized to function by normalizing the redox state of these cells by virtue of driving the LDH-A reaction [205,209]. Perhaps more importantly, because the exogenous pyruvate supply would now be uncoupled from NAD^+^-consuming upstream glycolytic reactions, excess NAD^+^ could also rescue the deficiency arising from mitochondrial dysfunction [210,211]. Using a Crispr/Cas9-based screen to identify genes necessary for maintaining proliferation in the face of mild, phenformin-induced Complex I dysfunction, Birsoy et al. [212] identified *GOT1*, which encodes cytosolic aspartate aminotransferase (GOT), the enzyme that catalyzes oxaloacetate’s conversion to aspartatre using glutamate as the amino donor. In doing so, GOT1 maintains the aspartate-malate shuttle by which cytoplasmic reducing equivalents are transferred into the mitochondrial matrix [213]. GOT1′s role was found to be a general one as its knockdown sensitized cells to other types of ETC dysfunction. Exogenous aspartate also rescued cells from the anti-proliferative effects resulting from GOT1 knockdown. Interestingly, unlike cells with a normal ETC, those with Complex I dysfunction generated aspartate via a third pathway involving glutamine’s reverse carboxylation to supply citrate and then the ATP citrate lyase-mediated conversion of citrate to oxaloacetate and acetyl-CoA in a GOT1-dependent manner. Thus, regardless of the means by which pyruvate rescued phenformin-mediate Complex I defects in WT cells, it required GOT1 and the apparent intactness of the aspartate-malate shuttle since it was unable to rescue *got1*-KO cells [212]. 

Cells with intact mitochondria but rendered Oxphos-defective due to a cytochrome B frameshift mutation are also pyruvate auxotrophs [214]. The notion that this was dependent on NAD^+^ regeneration was supported by the finding that α-ketobutyrate, which is also an LDH substrate, could replace pyruvate as an electron acceptor. Together, these results strongly suggested that pyruvate auxotrophy is a consequence of an Oxphos deficit and that exogenous pyruvate (or α-ketobutyrate) substitutes for oxygen as an electron acceptor.

α-ketobutyrate is oxidized to α-hydroxybutyrate and immediately excreted, thus explaining why it cannot maintain the viability of cells deprived of glucose and glutamine [214]. Unlike oxygen and α-ketobutyrate, which serve only as electron receptors, pyruvate’s support of proliferation in the absence of exogenous glucose indicated that it provided yet additional metabolic benefits. Sullivan et al. [214] found that non-proliferating cells with dysfunctional ETCs had subtle DNA synthesis defects, depleted stores of guanine 5′-monophosphate (GMP) and adenine 5′-monophosphate (AMP) and an excess of inosine 5′-monophosphate (IMP), a precursor of both GMP and AMP. The DNA replication-dependent proliferative defect could be partially rescued by adenine which, via a nucleotide salavage pathway, was converted to AMP. A clue to the nature of this apparent block lay in the fact that GMP synthesis is NAD^+^-dependent pathway whereas AMP synthesis depends on aspartate. Adenine’s incomplete ability to rescue these cells was therefore consistent with the fact that aspartate is needed for protein synthesis as well. Indeed, aspartate was better able to complement the proliferative defect as it allowed for both AMP and protein synthesis, while reducing the levels of IMP and increasing AMP but not GMP. As aspartate is not an electron acceptor and cannot normalize the NAD^+^/NADH ratio, these findings revealed that the rate-limiting step for proliferation in cells with defective ETCs was not the absence of electron acceptors as much as an absence of aspartate. The fact that pyruvate fulfills the role as both an electron acceptor and a precursor of aspartate while being non-inferior to aspartate at supporting proliferation, implied that a major function of Oxphos in proliferating cells was the synthesis of aspartate.

A question raised by the foregoing work is whether cells with an impaired ETC can regenerate NAD^+^ by means other than the LDH reaction. One way would be via gluconeogenesis which generates NAD^+^ during the conversion of 1,3-bisphosphoglycerate to glyceraldehyde 3-phosphate. Since pyruvate is an initial substrate for this pathway as well as for the LDH-A reaction, particularly when mitochondrial function is compromised, it would be of interest to determine whether gluconeogenesis is increased in these defective cells.

## 5. Malic Enzymes (MEs) 

### 5.1. The Three MEs Reside in Different Cellular Compartments and Have Different Co-Factor Requirements

The three MEs, namely ME1, ME2 and ME3, are oxidoreductases that catalyze the oxidative-decarboxylation of (S)-malate to pyruvate, CO_2_ and NAD(P)H [215,216]. ME1 is cytosolic and NADP^+^-dependent whereas ME2 and ME3 are mitochondrial and NAD(P)^+^-dependent [217,218]. ME1-3 link glycolysis and the TCA cycle and their activities are increased in many primary cancers and cancer cell lines [219,220,221,222,223,224,225,226]. The NADPH generated by ME1 also supports the rapid lipid biosynthesis associated with many tumors [172,227,228,229,230]. ME1 is also up-regulated by mutant forms of K-RAS and is suppressed by wild-type TP53. In NSCLC, ME1 levels correlate with radiation response [217,231,232].

### 5.2. ME1 Activity Is Regulated by Mutually Exclusive Phosphorylation and Acetylation of Adjacent Resides

Fernandes et al. [233] investigated ME1′s role in an animal model of intestinal tumorigenesis associated with the loss of the adenomatous polyposis coli (APC) tumor suppressor. Germ line *APC* mutations underlie the highly penetrant colorectal cancers (CRCs) associated with familial adenomatous polyposis (FAP) whereas acquired *APC* mutations are extremely common in sporadic human CRC [234,235]. Fernandes et al. [233] crossed Apc^Min/+^ mice to a strain that expressed ME1 in the intestinal epithelium and showed that the male offspring developed larger and more numerous intestinal adenomas. They also found that ME1 inhibition suppressed the in vitro growth of human CRC cells. However, these studies did not examine the metabolic consequences of ME1 over-expression or its regulation.

Enzymatically inactive monomeric ME1 reversibly assembles into active homo-dimers and homo-tetramers [236] (Figure 4). Zhu et al. identified two surface residues on ME1 as being critical for these transitions [237]. Phosphorylation of ME1 at Ser_336_ by the Ser/Thr kinase NEK1 blocked dimerization and enzymatic activation. This modification was reversed by the atypical Ser/Thr phosphatase PGAM5, a member of the phosphoglycerate mutase family that shuttles between the cytoplasm and mitochondria [238,239,240,241]. Due to altered charge, steric hindrance and/or conformational change, Ser_336_ de-phosphorylation permits ACAT1-mediated acetylation of the adjacent ME1 Lys_337_, thus licensing ME1 dimerization and activation. Ser_336_ phosphorylation and Lys_337_ acetylation were found to be mutually exclusive events with mutually exclusive consequences. Zhu et al. [237] further showed both ME1 and PGAM5 to be direct transcriptional targets of β-catenin, which is mutated and/or dysregulated in many cancers and is activated in FAP by virtue of APC mutational inactivation [242] (Figure 4). The question of how cytosolic ME1 could be a substrate for mitochondrial-localized ACAT1 was resolved by demonstrating that the ME1 phosphorylation mimetic S336D localizes to the mitochondria whereas the acetylation mimetic K337Q is cytosolic. The previous cytosolic assignment of ME1 [243] thus suggests that, under normal circumstances, it is largely inactive. It also explains the deacetylation of Lys_337_ by Sirt6, a cytosolic member of the sirtuin family [237].

The above results were confirmed and extended in a murine model of azoxymethane/dextran sulfate sodium (AOM/DSS)-induced CRC in which relevant ME1 mutants encoded by adenoviral vectors were over-expressed via intra-peritoneal delivery. Compared to mice injected with control vectors, those injected with vectors encoding wild-type ME1, phosphorylation-defective S336A or the acetylation-mimetic K337Q showed a marked increase in CRC number and size in association with increased NADPH production and lipid biosynthesis. Importantly, neither of the catalytically inactive ME1 S336D or S337R mutants altered CRC behaviors. The results confirmed the previous in vitro findings demonstrating the mutually exclusive nature of S_336_ and K_337_ post-translational modifications and their functional consequences.

In keeping with ME1 and PGAM5 being transcriptional targets for β-catenin, Zhu et al. [233] found β-catenin levels to be higher in tumor tissues and to correlate with ME1 and PGAM5 levels. NEK1 and Sirt6 were also expressed at lower levels in human CRC samples, which correlated with lower levels of ME1 Ser_336_ phosphorylation and higher levels of Lys_337_ acetylation. ME1 Ser_336_ phosphorylation and Lys_337_ acetylation were also inversely correlated. In turn, ME1 Ser_336_ phosphorylation directly correlated with NEK1 expression and Lys_337_ acetylation directly correlated with ACAT1 expression. Lastly, ME1 Lys_337_ acetylation correlated inversely with Sirt6 expression. These findings indicated that ME1 is subject to highly sensitive and coordinated positive and negative controls, mediated by mutually exclusive phosphorylation and acetylation reactions, which in turn dictate the oligomerization status of the protein, its subcellular localization and its overall activity. The tight regulation of ME1 activity and the readily reversible chemical reaction it catalyzes strongly suggest that ME1 and perhaps other ME members, act coordinately with enzymes such as PDC and PC to balance pyruvate, malate, oxaloacetate, and acetyl CoA in response to sometimes conflicting energy-generating and biosynthetic demands.

### 5.3. ME2 Supports Proliferation and Lipid Biosynthesis

Less information is available concerning the role of mitochondrial-localized and NAD(P)^+^-dependent ME2 although it is expressed at levels similar to those of ME1 and in many of the same tissues (https://www.genecards.org/, accessed on 1 February 2021). Activation also occurs via homotetramerization, is more substantial in established cancer lines and occurs early in transformation [223,244,245].

ME2 knockdown induced human K562 erythroleukemia cell differentiation, thus echoing previous observations made in Myc depleted cells [223,246,247]. Both the inhibition of Myc and the Myc-independent inhibition of ATP generation also induced terminal differentiation in HL60 promyelocytic leukemia cells and suggested that ATP depletion may be the common mechanism underlying proliferative arrest and differentiation in various cell types [223,224,246,247,248,249,250,251,252]. Extending their findings, Ren et al. found high ME2 transcript expression in a variety of tumor types, including 90% of lung cancers; in melanoma, ME2 levels correlated directly with disease stage [223].

Stable isotope-resolved metabolomics performed on ME2-knockdown A549 cells cultured in the presence of U-^13^C_6_-glucose or ^13^C_5_/^15^N_2_-glutamine showed reduced incorporation of glucose-derived label into membrane phospholipids. This suggested that, like ME1, ME2 also supports de novo lipid synthesis. This would almost certainly involve the generation of critical NADPH as discussed above as well as ensuring sufficient citrate synthesis for cytoplasmic diversion into acetyl-CoA synthesis. The increased demand for oxaloacetate, acetyl-CoA and citrate could readily be addressed by increasing PC or glutaminolysis [28,116,166,167,168,253,254,255]. Other biosynthetic pathways that utilize malate-derived pyruvate might benefit as well since the flexibility afforded by generating pyruvate from sources other than glucose would be of benefit during periods of glucose deprivation. Consistent with this, ME2 knockdown rendered pyruvate-, glutamine- and glucose-deprived A549 cells less able to be rescued by glutamine or to generate oxaloacetate via PC, presumably because they lacked sufficient supplies of ME2-generated pyruvate [223].

ME2 knockdown in other human cancer cell lines also reduced in vitro proliferation and colony formation and promoted a higher rate of spontaneous apoptosis and differentiation [224]. Surprisingly, the expected declines in ATP generation were accompanied by unexpected increases in oxygen consumption suggesting that these two processes were now uncoupled. Consistent with the redox imbalance that occurred in response to ME2 knockdown, the cells showed a higher NADP^+^:NADPH ratio, increased ROS production and a loss of anti-oxidant defenses.

Finally, the above group [223] speculated that the accumulation of malate in A549-ME2 knockdown cells might explain some of their phenotypes. Control A549 cells exposed to cell-permeable dimethyl malate increased ROS production, grew slowly, differentiated more robustly and eventually died. These features were mitigated by the addition of N-acetyl cysteine or glutathione indicating that ME2 depletion and/or malate accumulation direct these cellular changes indirectly via ROS induction. In vivo, both the constitutive and conditional knockdown of ME2 in A549 cells slowed tumor growth and increased their killing in response to cis-platinum treatment or the direct injection of dimethyl malate. The combination of cis-platinum and dimethyl malate also accelerated tumor differentiation.

Examination of the GEO Microarray data base by Cheng et al. [116] found ME2 transcripts to be modestly elevated in advanced gliomas (Grades III and IV [GBMs]). Functional consequences of ME2 knockdown in GBM cell lines (GBM8401 and LN229) included reduced growth rates, anchorage-independent clonogenicity, BrdU incorporation, matrigel-based migration and invasion. Although ATP levels were reduced in GBM8401 knockdown clones by only about 10%, ROS levels were decreased rather than increased as reported by Ren [223]. Inconsistent phosphorylation-dependent AMPK activation was observed along with no increases in p53 protein. The authors did not further document the in vivo behavior of these clones.

### 5.4. ME3 Is a Collateral Lethal Target in Pancreatic Cancer

ME3 has generally been less well-studied than ME1 and ME2. One interesting observation by Dey et al. [219] is that in pancreatic ductal adenocarcinoma (PDAC) homozygous deletion of the *SMAD4* gene within the 18q21 locus is quite common and often includes the neighboring *ME2* gene. This suggested that, because ME3 is also a mitochondrial enzyme, its function might be redundant with that of ME2 and thus be a potential synthetic lethal target in *ME2^−/−^* tumors or, in keeping with the co-deletion of *SMAD4*, a so-called “collateral lethal” target. Consistent with this notion, Dey at al [219] showed higher ME3 expression in PDAC cell lines lacking ME2 than in those with intact *ME2* loci and that ME3 knockdown was selectively toxic. They extended these findings to a number of additional primary PDAC and matched normal pancreatic tissue samples. Metabolic flux studies showed significantly increased conversion of [U-^13^C5]-glutamine into TCA cycle substrates in ME2/ME3-deficient PDAC cells. A noted increase in glutaminolysis may have represented a futile attempt to replenish the NADPH that could no longer be furnished by ME2 or ME3, given the fact that, in addition to NAD^+^, glutamate dehydrogenase can utilize NADP^+^ as a co-factor [256]. In parallel studies, [U-^13^C_6_]-glucose flux into lactate and pyruvate was unchanged although pyruvate’s entry into the TCA cycle was decreased and was accompanied by lower Oxphos, altered mitochondrial mass and structure, increased ROS and AMPK activation. Therefore, it was surprising that these cells were unable to compensate for the loss of mitochondrial function and ensuing energy deficit by increasing glycolysis.

Mitochondrial catabolism of branched-chain amino acids (BCAAs) was also profoundly down-regulated and correlated with reduced expression of BCAA Transaminase 2 (BCAT2), the mitochondrial enzyme responsible for the first step in BCAA metabolism by which α-ketoglutarate is transaminated to glutamate [215,219]. The mechanism by which this occurred required licensing by SHEBP1c, a sterol-regulated transcription factor whose AMPK-dependent phosphorylation blocked its translocation into the nucleus and its activation of the BCAT2 promoter. This seems somewhat counter-intuitive since a major function of AMPK is to replenish ATP deficits by up-regulating energy-generating processes such as BCAA oxidation [257]. On the other hand, the defects in mitochondrial structure and function, combined with the redox and ROS imbalance resulting from the dual loss of ME2 and ME3, might have caused a loss of coordination of this alternative means of acetyl-CoA generation. A similar inability to maintain ATP levels and mitochondrial structure occurs in *myc^−/−^* fibroblasts or in *myc^+/+^* cells whose mitochondria are maintained in a constant state of fission due to the over-expression of Drp1, a dynamin-like GTPase that participates in the terminal fission process [31,258]. Although the utilization β-FAO or acetate as alternate ATP sources was not tested by Dey et al. [219], it seems unlikely that ME2/ME3-deficient PDAC cells would be capable of overcoming their severe energy-generating restriction and redox imbalance.

### 5.5. Cooperation among the Three ME Isoforms

It is important to consider the possibility that, not only do MEs1–3 serve complimentary, synergistic and compartmentalized roles but do so in response to highly specific cell-type, proliferative and metabolic cues. For example, ME1 could play critical cytosolic roles by providing reducing equivalents for lipid and pentose sugar biosynthesis and redox balance while simultaneously furnishing pyruvate when glucose was limiting [259]. Glutamine anaplerosis could allow the “bottom half” of the TCA cycle to operate independently of acetyl-CoA generating malate and oxaloacetate via the standard TCA cycle reactions as well as malate-derived pyruvate via ME2 and ME3. PDC could then short-circuit glycolysis, replenish acetyl-CoA and restore the top portion of the TCA cycle. Additional glutamine-derived oxaloacetate could drive gluconeogenesis and provide glycolytically derived pyruvate.

Tumors tend to have highly oxidized intracellular environments due to ROS generated by dysregulated oncogenes such as Myc, inefficient electron transport, increased growth factor signaling and lactate over-production [31,207,260,261,262,263]. By providing cytoplasmic and mitochondrial NAD(P)H, ME1-3 could function cooperatively to counter redox imbalance by regenerating reduced forms of various cytosolic and mitochondrial redox-sensitive proteins including peroxidases, peroxiredoxin, glutathione reductase and thioredoxin [207,260,262].

Evidence for the differential regulation of all three ME isoforms was provided by Chang and Tong who analyzed their expression in multiple samples of normal skin, benign nevi and malignant melanomas [216]. ME2 transcripts and protein were increased in melanomas whereas ME1 and ME3 transcripts either decreased or remained unchanged. ME2 transcripts were particularly elevated in the acral lentiginous melanoma subclass that has high metastatic propensity. Ren et al. showed that ME2 knockdown in two melanoma cell lines reduced in vitro growth rates, anchorage-independent clonogenicity, in vivo xenograft growth and basal ATP levels while also activating AMPK [223,224]. A cell line with WT p53, but not one with mutant p53, up-regulated p53 and its target p21 leading to growth suppression, which could be rescued by re-expressing ME2. The p53 response, rather than being specifically related to the ME2 pathway, may have reflected a more general association between AMPK and p53, both of which respond to ATP deficits by reducing energy-intensive functions such as protein translation, DNA replication and cell division [264,265].

## 6. The Direct Generation of Acetate from Pyruvate

Despite its origination from sources other than pyruvate, mitochondrial acetyl-CoA supplies might remain low in the face of high proliferative demand, vascular insufficiency and nutrient scarcity and thus eventually become rate-limiting in tumors. Acetate, originating in the gut microbiome, or from intracellular deacetylation reactions, and converted to acetyl-CoA by the nucelo-cytoplasmic enzyme acetyl-CoA synthetase 2 (ACCS2) is necessary for the attainment of maximal tumor growth, particularly during hypoxia [11,266]. This could relieve the dependency on mitochondrial-derived acetyl- CoA and citrate that are the canonical starting substrates for de novo lipogenesis while conserving TCA cycle substrates for energy production [267,268,269].

In addition to the above sources, acetate is also derived from the diet and the hydrolysis of acetyl-CoA, [11,12,266,270,271]. However, evidence for its direct synthesis by eukaryotic cells has been largely indirect [132,272,273]. Liu et al. used liquid chromatography-high resolution MS to monitor the flux of uniformly labeled ^13^C-glucose in HCT116 CRC cells and several other cancer cell lines and found its rate of conversion to ^13^C_2_-acetate to rival that of its conversion to ^13^C_3_-pyruvate and ^13^C_3_-lactate [10]. Moreover, the amount of acetate generated exceeded by several orders of magnitude the amounts generated via deacetylation or hydrolysis reactions and <20% was derived from histone deacetylation. Inhibition of mitochondrial pyruvate transport minimally affected acetate levels, indicating that its origin was citrate-independent. Further ^18^O_2_ labeling studies showed that acetate was generated in response to a nucleophilic attack of pyruvate by ROS originating from H_2_O_2_ [10] An even more direct experiment performed under physiologically relevant conditions in vitro showed that pyruvate was non-enzymatically oxidized to acetate in the presence of H_2_O_2_ and was catalyzed by Cu^++^ at achievable intracellular concentrations. Finally, brief exposure of HCT116 cells to H_2_O_2_ increased their incorporation of ^18^O_2_ into acetate in a dose-dependent manner, whereas cells incubated to ^18^O_2_ and ^13^C_6_-glucose and exposed to H_2_O_2_ incorporated ^18^O into ~30% of newly synthesized, ^13^C-labeled, acetate.

The absence of ^18^O in a significant fraction of ^13^C-labeled acetate suggested the presence of an alternate, likely enzymatic, pathway of acetate generation. Liu et al. [10] thus focused on PDC given that its PDH moiety is also a keto acid decarboxylase. In a defined in vitro reaction containing purified PDC and all three of its co-factors, namely, thiamine pyrophosphate (TPP), CoA and NAD^+^, ^13^C_3_-pyruvate was converted to acetyl- CoA as expected, whereas in the presence of TPP alone, only ^13^C_2_-acetate and ^13^C_2_-acetaldehyde were generated. Liu et al. [10] additionally found the reaction to be catalyzed by α-ketoglutarate dehydrogenase, which, like PDH, is also a keto-acid dehydrogenase. The conversion of pyruvate to acetate by HCT116 cells was partially suppressed following the inactivation of PDH and even further suppressed in response to withholding thiamine, which is a cofactor for both PDH and α-ketoglutarate dehydrogenase. These findings were subsequently extended to mice bearing autochthonous sarcomas in which, following ^13^C_6_-glucose infusion, significant amounts of ^13^C_3_-pyruvate, ^13^C_3_-lactate and ^13^C_2_-acetate were detected in tumors and at levels much higher than those measured in serum.

Finally, Liu et al. showed that pyruvate-derived acetate synthesized by one cell could be transferred to and metabolized by another [10]. They showed that mouse embryo fibroblasts (MEFs) lacking ACLY, and unable to incorporate glucose-derived carbon into palmitate, were acetate auxotrophs. Co-culturing these with ^13^C_6_-glucose-labeled HCT116 cells sustained survival and the synthesis of lipids containing the ^13^C label originating from HCT116 cells. Moreover, knockdown of MEF ACCS2 blocked lipid synthesis indicating that the acetyl-CoA being used was derived from ^13^C_2_-acetate. Interestingly, ACLY inhibition in HCT116 cells had much less effect on lipid biosynthesis than it did in MEFs, whereas ACCS2 knockdown had a greater effect. This suggested that transformation affected the means by which cells gained access to and relied upon acetate.

Collectively, the above findings raise questions regarding the origin of the acetyl groups that decorate histones and other nuclear proteins. As some PDC and α-ketoglutarate dehydrogenase is also found in nuclei [10,85,90,274,275], it is tempting to speculate that acetate synthesized in the nucleus could be an immediately available substrate for enzymatically driven acetylation reactions [276]. The cellular “acetylome” is quite large, encompassing approximately 1000 distinct proteins some of which contain multiple acetylation sites [277]. It remains to be determined whether the acetate derived from different sources might localize to distinct cellular compartments and thus be utilized for different purposes.

## 7. Conclusions

This review has summarized current knowledge regarding how pyruvate is selectively distributed and utilized to maintain redox and metabolic balance while simultaneously matching energetic needs to growth rates. The means of achieving this involves six distinct but interactive and self-correcting pathways (Figure 1). As the end-stage product of glycolysis and the entry-level substrate of the TCA cycle, pyruvate occupies a unique and crucial link between aerobic and anaerobic metabolism. At the same time, its utilization by other pathways provides some assurance that glycolysis and the TCA cycle will be well-balanced so as to properly apportion their energetic, catabolic and anabolic duties in ways that do not restrict one another. Complicating these physiologic mandates is the fact that normal and neoplastic cells have distinct requirements that are addressed by re-programming these pathways in ways that are highly adaptable but whose frequent association with cancer qualifies them to be considered a major “hallmarks” [278]. At the same time, it is important to remember that these changes need not necessarily indicate reprogramming of the various enzymatic pathways per se but rather may also reflect altered levels of individual metabolites and reaction rates within pre-existing metabolic contexts [57]. Finally, the extracellular environment may not always be conducive to the support of maximal cellular metabolic health or growth. Such inimical conditions are common in cancer and pyruvate’s production and metabolism are central to initiating and sustaining this adaptive and flexible metabolism. Additional aspects of pyruvate metabolism will undoubtedly emerge in the near future and will surely add to the excitement and wonder over this versatile metabolite.

## Figures and Tables

**Figure 1 cells-10-00762-f001:**
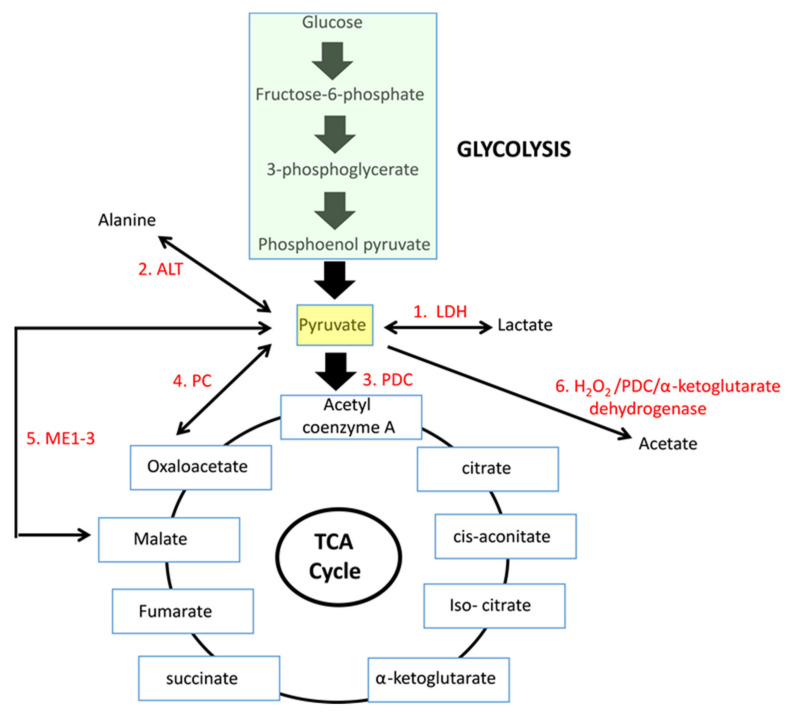
The six metabolic fates of pyruvate. The essentiality of pyruvate to central carbon metabolism is indicated (yellow box) 1. The LDH-mediated reduction to lactate; 2. The alanine amino transferase (ALT)-mediated transamination to alanine; 3. The PDC-mediated oxidation to acetyl- CoA; 4. The PC-mediated anaplerotic conversion to oxaloacetate; 5. The malic enzyme (ME)-mediated conversion to malate, and 6. The non-enzymatic generation of acetate via H_2_O_2_ or the enzymatic generation of acetate via PDH or α-ketoglutarate dehydrogenase.

**Figure 2 cells-10-00762-f002:**
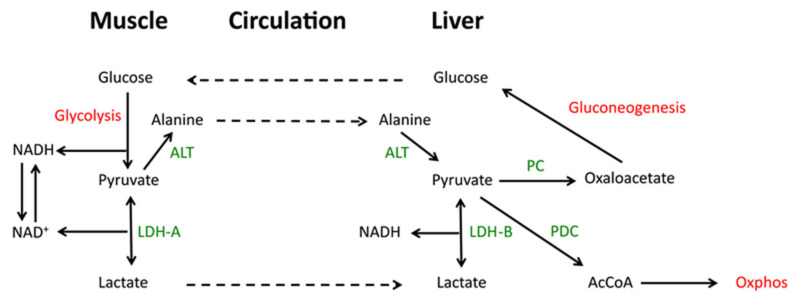
The central role of pyruvate in the Cori and alanine cycles. The Cori cycle: in muscle, anaerobic glycolysis favors lactate production and generates NAD^+^, which provides the necessary electron acceptor to support further glycolysis. Excreted lactate is converted to pyruvate in the liver via reversal of the LDH reaction. Pyruvate supports Oxphos following its conversion to acetyl-CoA by the PDC or gluconeogenesis following its PC-mediated conversion to oxaloacetate. Released glucose sustains plasma levels, which serve as an exogenous energy source. The alanine cycle: In muscle, alanine is derived from pyruvate via ALT or directly from the breakdown of proteins in response to fasting or tissue damage. The excreted alanine is then reconverted to pyruvate in the liver. Broken arrows: products released/excreted from cells.

**Figure 3 cells-10-00762-f003:**
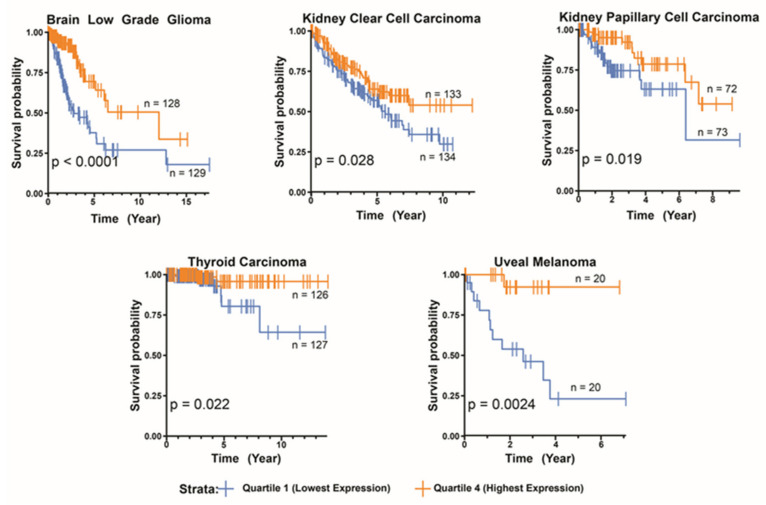
PC transcript levels in select tumor types correlate directly with survival. Normalized PC mRNA expression levels and Kaplan–Meier survival statistics were obtained from the TCGA PANCAN dataset available in the UCSC Xenabrowser for 10,593 samples across all cancer types. Within each cancer type, Kaplan–Meier survival differences between the top and bottom quartiles of PC expression were assessed using a log-rank test. Survival is expressed as the fraction of patients remaining alive from the time of diagnosis.

**Figure 4 cells-10-00762-f004:**
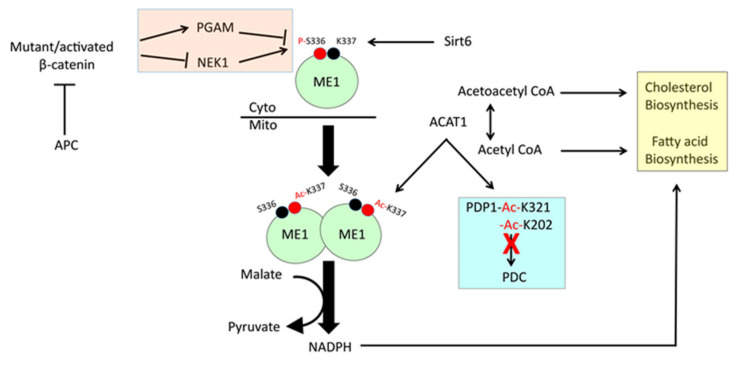
Regulation of ME1 by mutually exclusive phosphorylation and acetylation of adjacent residues. ME1 is normally maintained in an inactive, phosphorylated and non-acetylated cytoplasmic state via the balance between PGAM and NEK1 and the deacetylation activity of cytosolic Sirt6, respectively (post-translationally unmodified or modified residues are indicated by red and black small circles, respectively). Activation of β-catenin, either directly via mutation or indirectly via APC loss, alters the ratio between pro-oncogenic PGAM and tumor suppressor NEK1 so as to favor ME1^S336^ dephosphorylation, ME1′s translocation into the mitochondria and its ACAT1-mediated acetylation on K337. This in turn allows for ME1 homo-dimerization/tetramerization and the activation of its intrinsic enzymatic activity, which, while catalyzing the conversion of malate to pyruvate, generates NADPH that supports both fatty acid and cholesterol biosynthesis. ACAT1-mediated inhibitory acetylation of the PDP1 phosphatase inhibits PDC activity, allowing for the accumulation of pyruvate and other glycolytic intermediates, stimulating the Warburg effect and, in conjunction with increased lipid biosynthesis, promoting tumor growth.

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
