# Peer review of "The Metabolic Fates of Pyruvate in Normal and Neoplastic Cells"

_cells, 2021, doi:10.3390/cells10040762_

Round 1
Reviewer 1 Report
In their manuscript, E. V. Prochownik and H. Wang are interested in the fate of pyruvate in the metabolism of normal and neoplastic eukaryotic cells. Pyruvate is a key metabolite at the junction of anaerobic and aerobic metabolism that is finely tuned to meet the intricate requirements of both normal and tumor cells. The authors investigated the role of pyruvate through six distinct but interconnected metabolic pathways.
This is a very interesting and well documented review. This manuscript illustrates very well all the complexity of metabolism and highlights the many regulatory elements that allow the adjustment of the fluxes necessary for cell growth and homeostasis.
The manuscript should be suitable for publication but only after it has been revised to address the following points.
The manuscript is not so easy to read. The way it is presented, it may appear a bit like a series of examples of metabolic adaptations (as reported in the literature), all of which seem to be assessed with equal importance. Is this really the case, or are certain patterns more prevalent? From a synthetic point of view, the manuscript would certainly benefit from including subsections within each of the main chapters. Finally, in my opinion, the manuscript suffers from a lack of illustrations. Schematic illustrations for the main metabolic adaptations summarized in this review would make the reading of this work even more captivating.
Some minor comments:
Line 41: “it powers” seems to be a typo.
Figure 1: there is a typo : “6. H2O2/PDH/αKG gehydrogenase”.
Line 103: oxaloacetate is defined here as oxa for the first time. But oxa is mentioned in Figure 1, which is introduced earlier in the text (line 81).
Figure 2: there is a typo in oxaloacetate (“oxaloactate”). Alanine transferase is mentioned as ALT, but the term is not properly introduced in the main text.
Line 494: “MS” (mass spectrometry) is not introduced in the text.
Figure 3: I am curious about this figure. It seems to originate from an analysis made by the authors. How does this kind of analysis fit into a review work? The data presented are interesting but need to be more detailed. The figure also needs to be made more explicit. For example, what is the scale of the x-axis? How is the concept of time interpreted?
Line 690: there is a typo: “3C-labeled”.
Lines 839 and 861: there is a typo with “0-cells”.
Author Response
We were very pleased that the Reviewer considered our manuscript to be a “very interesting and well documented review [that] illustrates very well all the complexity of metabolism and highlights the many regulatory elements that allow the adjustment of the fluxes necessary for cell growth and homeostasis.” We have addressed the pertinent issues raised by the Reviewer in a point-by-point response below and have modified our revised manuscript accordingly.
Point 1: “The manuscript is not so easy to read…From a synthetic point of view, the manuscript would certainly benefit from including subsections within each of the main chapters.”
Response: We have now divided the review into subsections, have rearranged several other sections and have revised figures to better illustrate and emphasize certain pathways and concepts. We have also introduced a new figure (Fig. 4) that summarizes malic enzyme 1 regulation via mutually exclusive phosphorylation and acetylation reactions.
Point 2: Minor points
Response: All minor points have been addressed. Specifically, another reviewer asked that our abbreviations for oxaloacetate and a-ketoketoglutarate be written out in full.
Point 3: I am curious about this Figure” (Fig. 3).
Response: The Reviewer is correct in that the survival curves shown originated from an analysis performed by us specifically to illustrate our point for this article. The original data were extracted from The Cancer Genome Atlas (TCGA) compendium as stated in the text. Because TCGA contains the results of whole transcriptome RNAseq analyses performed on 34 different tumor types (>10,000 samples), it is possible to relate the expression of any transcript or group of transcripts to long-term survival. We have stated in the revised figure legend that “Survival is expressed as the fraction of patients remaining alive from the time of diagnosis.” This is the standard way of assessing long-term survival using standard Kaplan-Meier survival curves both for human as well as animal studies (1-4).
References
- Rich JT, Neely JG, Paniello RC, Voelker CC, Nussenbaum B, Wang EW. A practical guide to understanding Kaplan-Meier curves. Otolaryngol Head Neck Surg. 2010.143:331-6.
- Mandel J, Wang H, Normolle DP, Chen W, Yan Q, Lucas PC, Benos PV, Prochownik EV. Expression patterns of small numbers of transcripts from functionally-related pathways predict survival in multiple cancers. BMC Cancer. 2019.19:686.
- Mandel J, Avula R, Prochownik EV. Sequential analysis of transcript expression patterns improves survival prediction in multiple cancers. BMC Cancer. 2020.20:297.
- Zhang W, Meyfeldt J, Wang H, Kulkarni S, Lu J, Mandel JA, Marburger B, Liu Y, Gorka JE, Ranganathan S, Prochownik EV. β-Catenin mutations as determinants of hepatoblastoma phenotypes in mice. J Biol Chem. 2019.294:17524-17542.

Reviewer 2 Report
This manuscript presents a valuable review that can help the community to better circumvent the central role of pyruvate branch point in cell metabolic plasticity.
Although the list of cited references is of adequate size for a review, there are still some more to add such as to better explain Warburg effect. To the end, authors suggest in page 3 that Warburg effect occurs under oxygen limitation conditions, which is not accurate.
Figure 1 looks academic, and the concept to be shown may be better deserved showing central carbon metabolic pathway network with clearly indicating the six principal routes for pyruvate.
In Figure 2, showing the Cori and alanine cycle, reversible reactions must be shown between pyruvate and lactate, otherwise there is no cycles.
In Figure 3, results for only 5 cancer types are shown among the 34 analyzed by the authors. Although this selection may be the ones strongly corroborating authors' hypotheses, it may be useful to the community that all being shown. Survival probability should be plotted vs time in years, not in days. It is not clear also, from my reading, if these samples are coming from the same or different patients, or even dead patients.
Overall, a major improvement would be to add figures/tables to show/synthesize the state-of-knowledge on the main subjects exposed in the review manuscript, and the first one to add could present pathway regulation and regulatory mechanisms around the cell management through the pyruvate branch point.
Author Response
We were gratified that Reviewer 2 considered our manuscript to be a “valuable review that can help the community to better circumvent the central role of pyruvate branch point in cell metabolic plasticity.” We have addressed the issues raised by the Reviewer in a point-by-point response below and have modified our revised manuscript accordingly.
Point 1: “better explain Warburg effect. To the end, authors suggest in page 3 that Warburg effect occurs under oxygen limitation conditions, which is not accurate.”
Response: In the revised manuscript, we have corrected the statement implying that the Warburg effect is restricted to hypoxic environments.
Original section: Under these conditions, which may not necessarily provide sufficient oxygen to support Oxphos, glycolysis’ purpose is not only to support the energetic and biosynthetic needs associated with cell growth and division, but to do so anaerobically. This process, known as the Warburg effect, can function at rates that are not only sufficient to support these high biosynthetic demands but can provide enough pyruvate to support Oxphos as well when oxygen is plentiful [28,31-35].
Revised section (lines 107-110): This support of anabolic activity, which may occur under either hypoxic or normoxic conditions, is known as the Warburg effect. When necessary, it can support the high biosynthetic demands of proliferating cells yet still provide sufficient pyruvate to sustain oxygen-dependent Oxphos [28,31-38].
We also added three new references (1-3 below and 36-38 in the manuscript).
Point 2: “Figure 1 looks academic”
Response: We have re-designed Fig. 1 as suggested by the Reviewer and modified the Fig. legend to accommodate this change.
Point 3: “In Figure 2, showing the Cori and alanine cycle, reversible reactions must be shown between pyruvate and lactate, otherwise there is no cycles.”
Response: Thank you for pointing out this oversight, which has now been corrected
Point 4: “In Figure 3, results for only 5 cancer types are shown among the 34 analyzed by the authors.”
Response: The Reviewer is correct in that we identified five cancer types in which PC transcript levels correlated with survival. That this occur only with certain cancers is exceedingly common, indeed even the norm, and we have previously published numerous such examples (4-6). One example of this cited in the text (lines 819-821) concerns mutations in the SDHB subunit of Complex II of the ETC. These mutations are found almost exclusively in pheochromocytomas and paraganglioneuromas and, even in these cases, are confined to minority subsets of cases. Similarly, mutations in the APC gene are associated almost exclusively with sporadic or inherited colo-rectal cancers. Unless the Reviewer is insistent that we do so, we feel it is unnecessary to publish each of the instances where no such correlations were found. The method we originally used of presenting Kaplan-Meier survival results (x-axis represented in days) is very standard (4-6) but we modified the figures to express survival time time in years as suggested.
With regard to whether “these samples come from the same or different patients, or even dead patients”: the two survival groups in each panel (represented in blue and orange) represent separate cohorts (i.e. tumors containing the highest versus the lowest levels of transcripts. As is true for any Kaplan-Meier plot where differences in survival are being compared, the individuals comprising the two populations consist both of those who have died and those who remain alive (7). Kaplan-Meier statistics account for the fact that the populations under study are comprised of individuals that have usually been followed for different periods of time since diagnosis.
References
- Deberardinis RJ, Sayed N, Ditsworth D, Thompson CB. Brick by brick: metabolism and tumor cell growth. Curr Opin Genet Dev. 2008.18:54-61.
- Liberti MV, Locasale JW. The Warburg Effect: How Does it Benefit Cancer Cells? Trends Biochem Sci. 2016.41:211-218.
- Vaupel P, Schmidberger H, Mayer A. The Warburg effect: essential part of metabolic reprogramming and central contributor to cancer progression. Int J Radiat Biol. 2019.95:912-919.
- Mandel J, Avula R, Prochownik EV. Sequential analysis of transcript expression patterns improves survival prediction in multiple cancers. BMC Cancer. 2020.20:297.
- Mandel J, Wang H, Normolle DP, Chen W, Yan Q, Lucas PC, Benos PV, Prochownik EV. Expression patterns of small numbers of transcripts from functionally-related pathways predict survival in multiple cancers. BMC Cancer. 2019.19:686.
- Wang H, Lu J, Mandel JA, Zhang W, Schwalbe M, Gorka J, Liu Y, Marburger B, Wang J, Ranganathan S, Prochownik EV. Patient-Derived Mutant Forms of NFE2L2/NRF2 Drive Aggressive Murine Hepatoblastomas. Cell Mol Gastroenterol Hepatol. 2021.S2352-345X(21)00037-0.
- (https://en.wikipedia.org/wiki/Kaplan%E2%80%93Meier_estimator).

Reviewer 3 Report
Comments to manuscript Cells-1139515
According to the title, this manuscript set out to review the metabolic fates of pyruvate in normal and neoplastic cells. Pyruvate being a key metabolite at the crossroads to different metabolic pathways, the understanding of its contribution to the specific needs of the cell’s metabolism is fundamental for translational concepts in health and disease, including cancer. Thus this review brings in a timely contribution.
The manuscript is very comprehensive and informative, covering a wide range of references on the role of many enzymes and enzyme complexes involved in metabolism. It is basically very well written and well-structured in to different sections which have in focus different enzyme nodes embracing pyruvate as a substrate or product of six different pyruvate-associated metabolic pathways. Thus, this manuscript is not just about the metabolic fate of pyruvate as the title indicates, but goes in depth on the regulation and role of the pyruvate-engaging enzymes.
Critical comments:
- An appropriate title of this review should include the enzyme regulatory aspects, as it turns out to be the main emphasis of this manuscript.
- There is room for improvements regarding the organization and choice of the presented information within the chapters: The presentation of enzymes and enzyme complexes is not well balanced; some paragraphs get side-tracked in many details not pertinent to getting a clear picture of pyruvate’s metabolic role. Certainly, subheadings of the sections would guide the reader through the many aspects of the topic.
- Along this line, two aspects are only scantily and disparately alluded to with respect to the metabolic fate of pyruvate: For one, more words are needed on pyruvate uptake and release, i.e. the roles of plasma and mitochondrial membrane transporters (MCTs) (a brief mention in the Introduction, then in lines 784f); it deserves a section/subheading of its own. Secondly, LDHA and LDHB have different metabolic roles, which need to be differentiated more clearly in the section on LDH (it appears finally in lines 780f) Also, the role of alanine transferase in pyruvate metabolism is rather scanty.
- There is an overkill of information in some sections, which lets the reader lose sight of the context (cannot see the forest for the trees). And sometimes there are “holes” in the line of thinking when summarizing too much information in one very long sentence, especially when referring mainly to review articles. Too many of the references are indeed reviews, of which some are redundant; primary publications are more informative and provide access to the experimental data, allowing to recapitulate and reflect the conclusions.
- Sometimes, it is not clear whether intracellular or intercellular (e.g. between muscle and liver) pyruvate is meant. Moreover, the relevance of pyruvate in different cell types/tissues having different metabolic functions and phenotypes and should come out more clearly, as pyruvate will be engaged in different ways. Subheadings would help.
- Please, do write out the common metabolites by their full names: e.g. oxaloacetate, a-ketoglutarate; acetyl-CoA, et al. A list of abbreviations for the many different enzyme acronyms would be also very helpful in this review article
- It would be very helpful and make the discussed issues more transparent if the authors could provide graphs of the metabolic pathways of pyruvate around the enzymatic node being addressed in the specific sections.
- The section titles 4) ) and 6) should be reworded: An enzyme or enzyme complex is a structural component with a specific function, but is not a pathway in itself. An enzyme (complex) will constitute only one metabolite-converting reaction as part of a metabolic pathway.
- Also both sections 4), 5) and 6) read like a mini-review focused on the titled enzyme – with little reference to the role or fate of pyruvate. Some paragraphs read like a recapitulation of a single publication. In Section 4 for example, a recently published publication by the group (ref. 66) makes up for nine paragraphs (some without reference) in which its content is reiterated in quite some detail. Only in some cases does the relevance of the work to the metabolic fate of pyruvate then shine through. Similarly, in Section 6, several paragraphs report mainly on results from one publication. This is not my expectation of a overview review article.
- I suggest that the sections 4 (pyruvate dehydrogenase complex), 5 (pyruvate carboxylase) and 6 (malic enzyme) are relieved of too many (experimental) details not pertaining directly to the enzymes governing the metabolic fate of pyruvate, or then at least put pyruvate more in to perspective – to conform with the present title of the manuscript. These two sections are highly focused on the molecular biochemistry and regulation of enzyme complexes resulting from upstream growth factor-receptor kinases, hypoxia, knock-outs, etc. of these enzymes. Subtitles within these sections would help to find the conceptual way. Moreover, some paragraphs are dedicated to reporting on only one or two references (e.g. line 276f; 317f). A wrap-up on how these different components and conditions might affect pyruvate metabolism (scheme?) would give this paragraph a take-home message.
Some other specifics:
line 280: “....in accord with the idea that they were unable to generate sufficient pyruvate and other glycolytic intermediates to support Warburg respiration.” I do not understand this sentence. What is the basis for this idea? What does it mean: “Warburg respiration”? I could not find this term in PUBMED or other databases, nor is it defined in biochemical terms in the authors´ papers (e.g. ref. 61). Please avoid such ill-defined attributions of a biochemical effect.
line 298f.: This paragraph consists of a number of jumps in thinking. Explain the direct or indirect biochemical/regulatory connection between PKM2 and PDC. There too many things being implied which are not clear to the reader (e.g. “pool of pent-up Warburg intermediates”).
line 320: Please state the question (again).
line 362: A reference for steady-state profiling by MS is needed. This paragraph is missing some information. Where, and how does pyruvate accumulate in this context?
lines 609f: This is an example of sidetracking in to too many details: The paragraph highlights the role of PC and TCA-cycle in insulin production by beta-cells and only marginally discusses the metabolic fate of pyruvate.
lines 672f: How can it be that out of the high numbers of comparisons, a number of 5 examples is sufficient to conclude that high PC expression, correlating with favorable survival, is a surrogate for oxphos and reduced “Warburg-type respiration”? I do not understand this conclusion. Is this data published? What does it tell us about pyruvate metabolism?
lines 733f and lines 775f: As to the role of pyruvate in limiting nutrient conditions, a recent article shows changes in the fraction of glycolytic and TCA-replenished pyruvate as well as shifts in the engagement of ALT, PDH and PC by 13C-pyruvate.
Reference
Otto AM. Metabolic Constants and Plasticity of Cancer Cells in a Limiting Glucose and Glutamine Microenvironment—A Pyruvate Perspective. Front Oncol (2020) 10(2542). doi: 10.3389/fonc.2020.596197.
lines 784f: MCT1 and 4 also facilitate pyruvate transport. What is the balance between pyruvate and lactate transport? A note on quantitative relationships such as extracellular concentrations (ref 170) as well as KM-values would be meaningful in the pyruvate metabolic context.
Reference examples:
1) Jackson VN, Halestrap AP. The kinetics, substrate, and inhibitor specificity of the monocarboxylate (lactate) transporter of rat liver cells determined using the fluorescent intracellular pH indicator, 2',7'-bis(carboxyethyl)-5(6)-carboxyfluorescein. JBiolChem (1996) 271(2):861-8. 1
2) Contreras-Baeza Y, Sandoval P, Alarcón R, Galaz A, Cortés-Molina F, Alegría K, et al. Monocarboxylate transporter 4 (MCT4) is a high affinity transporter capable of exporting lactate in high-lactate microenvironments. J Biol Chem (2019) 294:20135-47. doi: 10.1074/jbc.RA119.009093.
line 893: Why, how is pyruvate the initial substrate for gluconeogenesis? Oxaloacetate is not only the product of the PC reaction, but could also come from malate and possibly aspartate.
line 1179f: What is the reference for the 18O2-studies? Three paragraphs report in some detail on the work von Liu et al. This could be shortened to the essential findings, as a reader interested in the details will likely study the original paper. (See also comments above.)
This manuscript requires substantial “digestive” processing of the overwhelming information, including reorganization and a more stringent selection of relevant material to serve the intended aim of the review. Moreover, it appears that not every reference is required, as some only indirectly endorse the messages of the text. Certainly, a review article cannot cover everything, but the reader expects an intelligent selection and reduction of the available information to get an overview of the known and unknown, data and concepts, of the field.
Therefore, I recommend a revision and compaction of the text, accompanied with schemes/graphs to reveal with more transparency the essences of the different metabolic pathways involved in pyruvate metabolism.
Author Response
We were pleased that Reviewer 3 considered this manuscript to be “very comprehensive and informative, covering a wide range of references on the role of many enzymes and enzyme complexes involved in metabolism. It is basically very well written and well-structured into different sections… Thus this review brings in a timely contribution.” We have addressed the issues raised by the Reviewer in a point-by-point response below and have modified our revised manuscript accordingly.
Point 1: “An appropriate title of this review should include the enzyme regulatory aspects, as it turns out to be the main emphasis of this manuscript.”
Response: Thank you for this helpful comment. We have retitled the manuscript “The Control of Pyruvate Metabolism in Normal and Neoplastic Cells” to better reflect the emphasis on the regulation of the pathways we discuss.
Point 2: “There is room for improvements regarding the organization and choice of the presented information within the chapters.”
Response: Reorganization and the inclusion of sub-sections was also recommended by Reviewer 1 and we have done so accordingly. Where possible, we have also eliminated certain more tangential aspects of the review. We believe this has resulted in the review being better organized and more readable.
Point 3: More information is needed on MCTs, LDH-A&B and ALT.
Response: Because the manuscript was already quite lengthy, we deliberately avoided a detailed discussion of its transport, especially in light of an excellent and very comprehensive review of MCTs published by Payen et al less than a year ago (1). However, at the request of the Reviewer, we lengthened this section a bit and referenced the Payen et al review. Particularly relevant to the main points of our review, we emphasized how MCT1 and MCT 4 complement one another: MCT 4 is preferentially used to transport lactate out of hypoxic (glycolytic) cells whereas MCT1 is preferentially used to transport extracellular lactate into cells where, following its conversion to lactate, it serves as a mitochondrial fuel (lines 976-986). We also have expanded somewhat on the differential roles of LDHA and LDHB. Other than what was already mentioned in the original manuscript, we were unable to locate very much additional information from the literature concerning ALT, its role in cancer or its regulation.
Point 4: “There is an overkill of information in some sections.”
Response: Please see our response to Comment 2. In the revised manuscript, we have tried to better consolidate, eliminate and reorganize some of the information so as to provide a more logical and less tangential flow.
Point 5: “Sometimes, it is not clear whether intracellular or intercellular (e.g. between muscle and liver) pyruvate is meant.”
Response: In the revised manuscript, we have now attempted to better indicate whether we are discussing intra- versus inter-cellular substrates. In general, if unstated, it should be assumed that we are referring to the former.
Point 6: Please, do write out the common metabolites by their full names: e.g. oxaloacetate, a-ketoglutarate; acetyl-CoA, et al. A list of abbreviations for the many different enzyme acronyms would be also very helpful in this review article.
Response: The full names for most substrates, particularly those used with some frequency (oxaloacetate, acetyl coenzyme A and a-ketoglutarate) are now written out fully and a list of abbreviations for other substrates, enzymes, etc. has been appended.
Points 7 & 8: “It would be very helpful and make the discussed issues more transparent… The section titles 4-6 should be reworded.”
Response: The term “pathway” has been removed from the titles for PC, PDC and Malic enzymes since, as correctly pointed out the Reviewer, they constitute single nodes rather the actual pathways.
Points 9 & 10: “ …sections 4-6 read like a mini-review focused on the titled enzyme.”
Response: At the request of the Reviewer, we have shortened the review and divided it into subsections as mentioned in response to other Reviewers’ comments. With regard to the section devoted to PDC, our original intention in reviewing pdh-/- fibroblasts in such detail was to emphasize the remarkable extent and magnitude of the resultant metabolic re-programming. This includes, for example, the reversal of the catabolic pathways for six-carbon sugars such as fructose and mannose, the accompanying excessive buildup of glycolytic substrates upstream of pyruvate and the altered response to oncogenic signaling by Myc. We believe this represents an important and under-appreciated consequence of the redirecting of metabolic intermediates that ensues in the wake of PDC loss but have tried to consolidate and shorten this and the other two sections.
Points 11-15: Re lines 280, 298, 320, 362, 602 (now lines 308, 326, 351, 394, 652)
Responses: Where possible sections have been re-written so as to be more comprehensible and to achieve a more logical flow with reduced “sidetracking”. We have also eliminated the use of the term “Warburg respiration’ to avoid confusion.
We have referenced the use of MS and better explained the basis of pyruvate’s accumulation (lines 401-404): “Increased CS activity, along with a plentiful supply of oxaloacetate provided by the PC reaction, might be expected to serve as an enzymatic “escape valve” to dispose of excess pyruvate that accumulates in the face of PDC inhibition and that might be particularly pronounced in the mitochondria, were CS, PC and PDC reside.”
Point 16: How can it be that out of the high numbers of comparisons, a number of 5 examples is sufficient to conclude that high PC expression, correlating with favorable survival, is a surrogate for oxphos and reduced “Warburg-type respiration”?
Response: This question is similar to that raised by Reviewer 2 (Point 4). Our findings were based on the well-established fact that increased glucose uptake by tumors is used clinically as a surrogate for the Warburg effect and often correlates with rapid tumor growth and poor prognosis (2-5). These features can occur in concert with the selective expression of the low-affinity PKM2 isoform of pyruvate kinase (5). We therefore speculated that the converse might be true in that high PC activity, by allowing the release of pent-up pyruvate, might represent a means to reduce glycolytic intermediates, minimize the Warburg effect and prolong survival as indeed we showed in Fig. 3, which has not been previously unpublished. We did not intend to imply that this finding is universal. In fact, in response to Reviewer 2, Point 4, we cited examples were dysregulation of cancer-associated genes is confined to only a single cancer or a small number of cancers, which is quite typical. We also provided contrary examples (7-9). Recent findings, including those from our own group, have shown that dietary interventions such as high-fat diets can reverse tumor glycolysis, increase oxphos and prolong survival (10,11). This indicates that such metabolic rewiring is not irreversible but rather is highly adaptable and opportunistic and, as such, can be manipulated to achieve clinically desired results. We believe the re-writing of this section has helped to clarify our point. However, we did state in the manuscript that this speculation was only one possible explanation for this finding.
Point 17: lines 733f and lines 775f: As to the role of pyruvate in limiting nutrient conditions, a recent article shows changes in the fraction of glycolytic and TCA-replenished pyruvate as well as shifts in the engagement of ALT, PDH and PC by 13C-pyruvate.
Response: Thank you for calling our attention to this article (now included as ref. 56). While the cited studies were performed in vitro and with only a single cell line, they do support our contention that PC and glutamine anaplerotic flux tend to be mutually exclusive. Perhaps more importantly (and a point that we tended to downplay in our original iteration), this paper emphasized that so-called metabolic reprogramming may sometimes merely reflect the dynamic changes in metabolite quantity, reaction rates, and directions of the existing metabolic network rather than an actual physical re-programming. Of course, these adaptations are not mutually exclusive. We now refer to this article at several points and also mention it in the Conclusion of our revised manuscript.
Point 18: “lines 784f: MCT1 and 4 also facilitate pyruvate transport.”
Response: The Reviewer raises an interesting point that MCT1 and MCT 4, in addition to transporting lactate, can also transport pyruvate. In the case of MCT1 the Km(lactate):Km(pyruvate) is ~3.6 whereas for MCT4-undoubtedly the more important of the two in cancer-the ratio is ~0.4 (12). Moreover, in normal tissues and serum the lactate concentration is typically ~10-fold higher than the pyruvate concentration and increases as much as 20-fold more in highly glycolytic tumors (13-15). MCT4 is often markedly up-regulated in cancer and can therefore efficiently provide cancer cells with an atypical fuel (i.e. lactate) (1). Although not directly germane to the point to which the Reviewer refers (line 784, now line 972), we have included mention that the transport of lactate into tumor cells is more efficient, particularly in the face of MCT4 up-regulation.
Point 19: “line 893: how is pyruvate the initial substrate for gluconeogenesis? Oxaloacetate is not only the product of the PC reaction, but could also come from malate and possibly aspartate.”
Response: The reviewer is correct in that pyruvate is not the sole source of oxaloacetate. As we discuss at other points in the manuscript, it can originate elsewhere, such as from increased glutaminolysis, from malic enzyme or from the catabolism of aspartate. However, this paragraph speculated upon the source of oxaloacetate when Oxphos was compromised and mitochondrial supplies were low and glycolytic sources (e.g. pyruvate) were high. This point has been addressed by changing the sentence “Since pyruvate is the initial substrate for this pathway….” to “Since pyruvate is an initial substrate for this pathway as well as for the LDH-A reaction, particularly when mitochondrial function is compromised…” (line 916).
Point 20: “line 1179f: What is the reference for the 18O2-studies?”
Response: The reference for the 18O2-labeling studies has been added and we have attempted to consolidate this section
References
- Payen VL, Mina E, Van Hée VF, Porporato PE, Sonveaux P. Monocarboxylate transporters in cancer. Mol Metab. 2020 Mar;33:48-66.
- Gatenby RA, Gillies RJ. Why do cancers have high aerobic glycolysis? Nat Rev Cancer. 2004.4:891-9.
- Kunkel M, Reichert TE, Benz P, Lehr HA, Jeong JH, Wieand S, Bartenstein P, Wagner W, Whiteside TL. Overexpression of Glut-1 and increased glucose metabolism in tumors are associated with a poor prognosis in patients with oral squamous cell carcinoma. Cancer. 2003.97:1015-24.
- Mochiki E, Kuwano H, Katoh H, Asao T, Oriuchi N, Endo K. Evaluation of 18F-2-deoxy-2-fluoro-D-glucose positron emission tomography for gastric cancer. World J Surg. 2004.28:247-53.
- Park AK, Lee JY, Cheong H, Ramaswamy V, Park SH, Kool M, Phi JH, Choi SA, Cavalli F, Taylor MD, Kim SK. Subgroup-specific prognostic signaling and metabolic pathways in pediatric medulloblastoma. BMC Cancer. 2019.19:571.
- Dayton TL, Jacks T, Vander Heiden MG. PKM2, cancer metabolism, and the road ahead. EMBO Rep. 2016.17:1721-1730.
- Delgado-Goñi T, Galobart TC, Wantuch S, Normantaite D, Leach MO, Whittaker SR, Beloueche-Babari M. Increased inflammatory lipid metabolism and anaplerotic mitochondrial activation follow acquired resistance to vemurafenib in BRAF-mutant melanoma cells. Br J Cancer. 2020.122:72-81.
- Phannasil P, Thuwajit C, Warnnissorn M, Wallace JC, MacDonald MJ, Jitrapakdee S. Pyruvate Carboxylase Is Up-Regulated in Breast Cancer and Essential to Support Growth and Invasion of MDA-MB-231 Cells. PLoS One. 2015 Jun12;10(6):e0129848.
- Shinde A, Wilmanski T, Chen H, Teegarden D, Wendt MK. Pyruvate carboxylase supports the pulmonary tropism of metastatic breast cancer. Breast Cancer Res. 2018.20:76.
- Kanarek N, Petrova B, Sabatini DM. Dietary modifications for enhanced cancer therapy. Nature. 2020.579:507-517.
- Wang H, Lu J, Dolezal J, Kulkarni S, Zhang W, Chen A, Gorka J, Mandel JA, Prochownik EV. Inhibition of hepatocellular carcinoma by metabolic normalization. PLoS One. 2019. 14:e0218186.
- Contreras-Baeza Y, Sandoval PY, Alarcón R, Galaz A, Cortés-Molina F, Alegría K, Baeza-Lehnert F, Arce-Molina R, Guequén A, Flores CA, San Martín A, Barros LF. Monocarboxylate transporter 4 (MCT4) is a high affinity transporter capable of exporting lactate in high-lactate microenvironments. J Biol Chem. 2019.294:20135-20147.
- Go S, Kramer TT, Verhoeven AJ, Oude Elferink RPJ, Chang JC. The extracellular lactate-to-pyruvate ratio modulates the sensitivity to oxidative stress-induced apoptosis via the cytosolic NADH/NAD+redox state. Apoptosis. 2021.26:38-51.
- Walenta S, Wetterling M, Lehrke M, Schwickert G, Sundfør K, Rofstad EK, Mueller-Klieser W. High lactate levels predict likelihood of metastases, tumor recurrence, and restricted patient survival in human cervical cancers. Cancer Res. 2000.60:916-21.
- Otto AM. Metabolic Constants and Plasticity of Cancer Cells in a Limiting Glucose and Glutamine Microenvironment-A Pyruvate Perspective. Front Oncol. 2020 Dec 8;10:596197.
Round 2
Reviewer 1 Report
The authors have addressed the various points. The manuscript can now be accepted in the present form.
Author Response
Thanks
Reviewer 3 Report
Comments to revised manuscript Cells 1139515
The manuscript has been well revised in many parts, which certainly makes this review article more rounded up, transparent and readable. I highly appreciate the responses and improvements by the authors
However, some of the remarked improvements do not yet have the necessary impact in the revised manuscript, in particular those with respect to the subheadings and the condensation /trimming of long self-referenced paragraphs in sections 4 and 5.
A number of issues still needing attention are indicated below.
- The new title is fine, but it was not revised in the pdf-file.
- The manuscript is structurally not yet well set. The subheadings are sometimes overloaded and some additional ones would be useful (especially in section 4 and 5). These should serve as an outline in such an extensive article. Presently, they do not follow a logical and common format (see Prentice Hall Handbook for Writers). I suggest using short phrases to indicate the topic /subsection of the corresponding paragraph(s) – not long wordy sentences, as for example, in subheading 3.1, or 4.5.
- Another suggestion: The Introduction could actually incorporate section: 2. General features of pyruvate generation from glucose.
- line 111: This sentence is not quite correct: How does entry of pyruvate in to the TCA cycle “initiate” oxphos, as the respiratory chain reactions are several steps off and can be fed by other downstream metabolites?
- In Figure 1: the arrow in the TCA cycle should be removed – as most reactions operate in both directions.
- Line 129: Heading 3: the title needs revision – either naming both enzymes, or the two different reactions for pyruvate conversion.
- line 131: LDH is mostly cytoplasmic, but it has also been found in mitochondria (see also section 3.2 on gluconeogenesis).
(Hussien R, Brooks GA. Mitochondrial and Plasma Membrane Lactate Transporter and Lactate Dehydrogenase Isoform Expression In Breast Cancer Cell Lines. Physiol Genomics (2011) 43(5):255-64.
The lines 143f now provide important information on the diverse roles of LDH and round up the topic more comprehensibly. Also, the sentences on the role of MCTs now amend the topic of pyruvate/lactate uptake.
- In section 4, line 239: “Warburg respiration” is still in the text – do rephrase to a biochemical meaningful term.
- Line 258: Please translate “pyruvate contributes to the anabolic objectives of the Warburg effect” into biochemical terms.
- line 260f: I do not understand this sentence: how does a failure in PKM2 up-regulation lead to a loss of PEP? This statement is not supported by the ref. 57. Do you mean PCK2 up-regulation?
- Section 4 .5 is very lengthy (4 pages), basically un-compacted, and it still lacks thoughtful (subheading) guidance. It reads like an extensive report/summary on the authors’ published work with a specific experimental model system using genetically modified hepatocytes and hepatoma as well as fibroblasts. For readers of a review article, who may not be specialists in the field and who are firstly interested in getting an overview of the state of the field and a gain of new knowledge, it would be more useful to serve this expectation. Of course, the experimental concept and system should be mentioned, but reporting too many details and observations overloads the text and blurs the conclusions. A metabolic scheme would emphasize the importance and improve the impact of this section. Those readers interested in the details will confer the references.
- Section 5 reads much better now. But in some parts of Section 5 (namely 5.2 and 5.3), the comments of point 11. also apply.
- In Figure 1 and line 544 it is suggested that the PC reaction is reversible. This claim is not found in the cited references nor in medical biochemistry textbooks, nor is it discussed here in the context of the metabolic network. What is the reference for this statement?
- Line 911: “... the major function of oxphos in proliferating cells was the synthesis of aspartate...”. I find this to be a questionable implication, even in the context.
- More figures providing reaction schemes to the topic of the individual sections (like Fig.4, for example) would certainly help guide the reader through the labyrinth of metabolic pathways being discussed.
..... - Some editorial comments:
- Please start a paragraph with a sentence introducing the topic – not with the first-author of a reference (see lines 180, 456, 531,740, 766, ... other paragraphs)
- In biochemistry text books and publications, it is usage to write “acetyl-CoA” rather than spelling out “coenzyme.”
- line 579: the citation text should be removed.
- Some sentences need commas to avoid misreading, e.g. 546; 547; et al.
- Please add to the legend of Fig.4 the labeling code for the black and red dots.
While there is a wealth of information in this manuscript nicely consolidated, with most parts reading smoothly and coherently, some parts are yet with bulky text (especially section 4), not well structured, too much content and details, sometimes being redundant, sometimes distracting from the essential messages.
Therefore, in the present state of this otherwise interesting overview, I recommend further revisions.
Author Response
RESPONSE TO REVIEWER 3
To the editors: We have responded below to the second set of comment provided by Reviewer 3, which were the only ones provided.
Point 1: “The new title is fine, but it was not revised in the pdf-file.”
Response: We believe this was an oversight of the Editorial Office that should now be rectified.
Point 2: “The manuscript is structurally not yet well set…The subheadings are sometimes overloaded”
Response: We attempted to summarize the content of each subsection with these subtitles and were thus a bit too verbose in doing do. We have shortened some these as suggested, hopefully without losing the most important point (please note especially sections 3.1 and 4.5, now 2.1 and 3.5, respectively):
Previous: 3.1. LDH supports glycolysis, serves as an alternative fuel and can provide the basic substrate for gluconeogenesis.
New: 2.1. LDH supports glycolysis and provides an alternative fuel source.
Previous: 5.2. Anaplerotically-derived oxaloacetate contributes to multiple, non-mitochondrial anabolic pathways in both normal and tumor cells.
New: 4.2. Anaplerotically-derived oxaloacetate contributes to multiple, non-mitochondrial anabolic pathways.
Previous: 4.3: The contribution of oncogenic signaling to PDC regulation.
New: 3.3: Oncogenic signaling regulates PDC
Previous: 4.5 Disrupting PDH has surprisingly little effect on the proliferation of normal and transformed cells but is associated with significant compensatory metabolic reprogramming.
New: 3.5. Disrupting PDH has surprisingly little effect on proliferation.
Previous 5.1. The PC reaction is a major anaplerotic source of oxaloacetate for the TCA cycle.
New 4.1. PC is a major anaplerotic source of oxaloacetate
Point 3: “The Introduction could actually incorporate section: 2. General features of pyruvate generation from glucose.”
Response: Section 2 has now been incorporated into the Introduction as suggested
Point 4: line 111: This sentence is not quite correct:
Response: Previous sentence: “Finally, glycolytically-derived pyruvate can enter the TCA cycle to initiate Oxphos and support additional biosynthetic need”
New sentence: “Finally, glycolytically-derived pyruvate can enter the TCA cycle to support both Oxphos and additional biosynthetic needs.”
Point 5: In Figure 1: the arrow in the TCA cycle should be removed – as most reactions operate in both directions.
Response: The arrow has been removed from the figure.
Point 6: Line 129: Heading 3: the title needs revision – either naming both enzymes, or the two different reactions for pyruvate conversion.
Response: This has been revised.
Previous: 3. The Lactate Dehydrogenase (LDH) and Alanine Biosynthetic Pathways
New: 2. The Lactate Dehydrogenase (LDH) Reaction and Alanine Biosynthetic Pathway
Point 7: line 131: LDH is mostly cytoplasmic, but it has also been found in mitochondria (see also section 3.2 on gluconeogenesis).
Response: This reference has been included in the revised manuscript and the line in question has been altered to indicate that LDH may sometimes be mitochondrial.
Point 8: In section 4, line 239: “Warburg respiration” is still in the text – do rephrase to a biochemical meaningful term.
Response: “Warburg respiration” has been changed to “the Warburg effect”
Point 9: Line 258: Please translate “pyruvate contributes to the anabolic objectives of the Warburg effect…” into biochemical terms.
Response: We are a bit confused by this point. The entire sentence reads: “…pyruvate contributes to the anabolic objectives of the Warburg effect by serving as the initial biosynthetic substrate for alanine and other amino acids while providing the essential anaplerotic substrate oxaloacetate (see below).” We believe it is framed in appropriate biochemical terms. However, we did alter it a bit to make it less ambiguous: “…pyruvate contributes to the anabolic objectives of the Warburg effect by serving as the initial biosynthetic substrate for alanine and other amino acids and by providing the essential anaplerotic substrate oxaloacetate. This further supports aspartate biosynthesis and gluconeogenesis (see below).”
Point 10: line 260f: I do not understand this sentence: how does a failure in PKM2 up-regulation lead to a loss of PEP? This statement is not supported by the ref. 57. Do you mean PCK2 up-regulation?
Response: PKM2 is a low-affinity isoform of PKM that, in cancer, may replace the higher-affinity PKM1 isoform of normal cells. As a result, PEP, which lies immediately upstream, will accumulate. Failing to replace PKM1 by PKM2 would prevent this as well as the accumulation of other upstream glycolytic substrates and thus temper the Warburg effect. In retrospect, the sentence as written, while correct, is convoluted and has now been re-phrased: “Finally, the failure to replace PKM1 with PKM2 may not necessarily limit the supply of PEP; it may still be generated from oxaloacetate via the proximal portion of the gluconeogenic pathway, particularly if PDC activity were reduced.”
Points 11&12: Section(s) 4 (and) 5 are very lengthy (4 pages), basically un-compacted, and it still lacks thoughtful (subheading) guidance.
Response: We have shortened these sections by >20% by removing some of the more granular experimental details and less relevant discussions, while providing additional subheadings. The section on pyruvate carboxylase (now section 4) has also been rearranged to provide better flow.
Point 13: The PC reaction is irreversible.
Response: Lehninger’s 1970 first edition of “Biochemistry” (p.355) and the original work of Utter and colleagues, who first purified and characterized PC, describe the reaction as being reversible (DG = -0.5 kcal) (for example, see Scrutton MC and Utter MF. Pyruvate carboxylase. 3. Some physical and chemical properties of the highly purified enzyme. J Biol Chem. 1965.240:1-9, as well as other papers in this series). The latter reference has been added to the manuscript to support this claim. It is not clear how the PC reaction came to be considered as irreversible. However, we suspect it stems from the paper cited above, which illustrates the reaction as being uni-direction while describing it in the text as being quite reversible. Subsequent papers from the Utter group also present the reaction as being reversible (for example: Mildvan AS et al. Pyruvate carboxylase. VII. A possible role for tightly bound manganese. J Biol Chem. 1966.241:3488-98). We suspect the idea of PC irreversibility may have been propagated as a result of this early misrepresentation.
Point 14: Line 911: “... the major function of oxphos in proliferating cells was the synthesis of aspartate...” I find this to be a questionable implication, even in the context.
Response: It is not clear to us what the reviewer is suggesting. Our intention-in keeping with the spirit of the review to describe the metabolic fates of pyruvate-was to emphasize the fact that pyruvate plays a unique role in metabolism by serving as both an electron acceptor and a precursor of aspartate. The statement that “…the major function of Oxphos in proliferating cells was the synthesis of aspartate rather than the regeneration of electron acceptors.” was a paraphrasing of the conclusions drawn in the cited paper by Sullivan et al (Cell. 2015.162:552-63: “Nevertheless, exogenous aspartate addition is sufficient to restore proliferation of cells that otherwise stop proliferating or die when ETC activity is impaired. Thus, a primary role for mitochondrial respiration in cell proliferation must be to provide access to electron acceptors in support of aspartate synthesis.”). We have altered the wording of our sentence which could be mis-interpreted as indicating that the provision of aspartate is the sole purpose of the mitochondria.
Point 16: All minor editorial modifications have been made.
Finally, we believe the reorganization and deletion of some of the more tangential sections as mentioned above has reduced the bulkiness of the review in general and provided for a more readable and less confusing summary.
We are particularly indebted to Reviewer 3 for taking the time to thoroughly evaluate this manuscript and make numerous suggestions that we believe substantially enhance its readability.